# Insulin mutations impair beta-cell development in a patient-derived iPSC model of neonatal diabetes

**Diego Balboa[1]\*, Jonna Saarimäki-Vire[1], Daniel Borshagovski[2], Mantas Survila[2], Päivi Lindholm[3], Emilia Galli[3], Solja Eurola[1], Jarkko Ustinov[1], Heli Grym[1], Hanna Huopio[4], Juha Partanen[2], Kirmo Wartiovaara[1,5], Timo Otonkoski[1,6]\***

[1]Research Programs Unit, Molecular Neurology and Biomedicum Stem Cell Centre, Faculty of Medicine, University of Helsinki, Helsinki, Finland; [2]Department of Biosciences, University of Helsinki, Helsinki, Finland; [3]Institute of Biotechnology, Helsinki Institute of Life Science, University of Helsinki, Helsinki, Finland; [4]University of Eastern Finland and Kuopio University Hospital, Kuopio, Finland; [5]Clinical Genetics, HUSLAB, Helsinki University Central Hospital, Helsinki, Finland; [6]Children's Hospital, University of Helsinki and Helsinki University Hospital, Helsinki, Finland

**\*For correspondence:**
diego.balboa@helsinki.fi (DB);
timo.otonkoski@helsinki.fi (TO)

**Competing interests:** The authors declare that no competing interests exist.

**Abstract** Insulin gene mutations are a leading cause of neonatal diabetes. They can lead to proinsulin misfolding and its retention in endoplasmic reticulum (ER). This results in increased ER-stress suggested to trigger beta-cell apoptosis. In humans, the mechanisms underlying beta-cell failure remain unclear. Here we show that misfolded proinsulin impairs developing beta-cell proliferation without increasing apoptosis. We generated induced pluripotent stem cells (iPSCs) from people carrying insulin (*INS*) mutations, engineered isogenic CRISPR-Cas9 mutation-corrected lines and differentiated them to beta-like cells. Single-cell RNA-sequencing analysis showed increased ER-stress and reduced proliferation in INS-mutant beta-like cells compared with corrected controls. Upon transplantation into mice, INS-mutant grafts presented reduced insulin secretion and aggravated ER-stress. Cell size, mTORC1 signaling, and respiratory chain subunits expression were all reduced in *INS*-mutant beta-like cells, yet apoptosis was not increased at any stage. Our results demonstrate that neonatal diabetes-associated INS-mutations lead to defective beta-cell mass expansion, contributing to diabetes development.
DOI: https://doi.org/10.7554/eLife.38519.001

## Introduction

Pancreatic beta-cells maintain glucose homeostasis *via* the regulated secretion of insulin. Although the etiologies of type 1, type 2 and monogenic diabetes are different, they share similarities in the molecular pathways that become dysregulated in beta-cells during disease progression. Among these, endoplasmic reticulum (ER) stress and unfolded protein response (UPR) seem to be critical for the proper function and resilience of the beta-cell, and their role has been studied in different diabetes models (*Brozzi and Eizirik, 2016*; *Cnop et al., 2017*; *Herbert and Laybutt, 2016*). High quantities of insulin are transcribed, translated and ultimately secreted by beta-cells. This requires the establishment of appropriate mechanisms for proinsulin translation, folding, processing, storage and eventual secretion of mature insulin (*Steiner et al., 2009*). To cope with both the constant basal insulin secretion and the dynamic demand in response to elevated circulating glucose, the UPR is highly efficient in beta-cells, and adapts the ER loading and protein folding capacity to the insulin biosynthesis rate (*Back and Kaufman, 2012*; *Vander Mierde et al., 2007*). High levels of insulin

**eLife digest** Insulin is a hormone that is crucial for maintaining normal blood sugar levels and is produced by so called beta cells in the pancreas. If the beta cells in the body stop making insulin, blood sugar levels start to rise, which can lead to diabetes. A form of diabetes known as neonatal diabetes, where the body stops making insulin, usually appears during the first six months of life.

Infants affected by this early onset of diabetes often have mutations in one copy of the gene that encodes insulin. This means that they can still produce half of the amount of insulin, but it is not enough to keep blood sugar stable. Instead, insulin production stops completely after a few months. Scientists believe that this is because the mutant insulin has a toxic effect on beta cells.

Mutations in the insulin gene can affect the structure of insulin. As a result, insulin accumulates inside the beta cells, which stresses them and eventually makes them fail. The mechanisms behind this process are still unclear. Now, Balboa et al. used stem cells (which can turn into other cell types) taken from patients with this rare type of insulin mutation to find out more.

They corrected the mutant insulin gene in these stem cells with a technique called CRISPR and then induced the mutant and corrected stem cells to turn into beta cells. The results showed that the mutant beta cells slowed down their rate of cell division but did not die more frequently. When the cells were implanted into mice their growth and development changed. The mutant cells were more stressed and smaller than the cells with the repaired genes. They also had fewer signalling molecules that help cells grow. As a consequence, the cells were struggling to grow and mature.

Although this type of diabetes is rare, beta cells come under stress in other forms of the disease. In a separate study, Riahi et al. found that boosting molecular signals for cell growth could protect beta cells in mice with mutant insulin. If this could also work in humans, it may lead to new ways to prevent diabetes.

DOI: https://doi.org/10.7554/eLife.38519.002

biosynthesis generate a chronic sub-threshold ER-stress that suppresses beta-cell proliferation (*Szabat et al., 2016*), while induction of mild ER-stress in the context of hyperglycemia has been shown to induce beta-cell proliferation (*Sharma et al., 2015*). These findings highlight the important link between insulin expression, UPR levels and beta-cell proliferation.

Permanent neonatal diabetes mellitus (PNDM) is caused by mutations in genes controlling beta-cell development or functionality, and is usually diagnosed before 6 months of age (*Greeley et al., 2011*; *Murphy et al., 2008*). The development of efficient differentiation protocols has enabled the generation of beta-like cells in vitro from human pluripotent stem cells (hPSC) (*Pagliuca et al., 2014*; *Rezania et al., 2014*; *Russ et al., 2015*). Combined with genome editing technologies, they make possible the establishment of in vitro models for detailed studies of pathogenic mechanisms of PNDM (*Balboa and Otonkoski, 2015*; *Saarimäki-Vire et al., 2017*; *Shang et al., 2014*; *Zhu et al., 2016*). Insulin gene mutations are among the most common causes for PNDM globally (*Huopio et al., 2016*; *Støy et al., 2010*). Dominant negative heterozygous mutations that disrupt cysteine bridges within proinsulin lead to its misfolding, aggregation and accumulation in the ER (*Herbach et al., 2007*; *Liu et al., 2010a*; *Park et al., 2010*; *Rajan et al., 2010*). Accordingly, these high molecular weight proinsulin aggregates increase ER-stress and activate the UPR. Sustained UPR activation results in beta-cell dysfunction and the eventual onset of diabetes (*Colombo et al., 2008*; *Liu et al., 2010b*). This phenomenon has been studied extensively in the Akita mouse model of diabetes, which carries a proinsulin cysteine disruption mutation (C96Y) that leads to mutant proinsulin accumulation in the ER, enlarged ER, reduction of secretory granules and mitochondrial swelling (*Izumi et al., 2003*; *Kayo and Koizumi, 1998*; *Wang et al., 1999*; *Yoshioka et al., 1997*; *Zuber et al., 2004*). Similar findings have been reported from the Munich mouse model carrying *Ins2* C95S mutation (*Herbach et al., 2007*). Although further studies suggested that unresolved UPR resulted in beta-cell apoptosis *via Chop (Ddit3)* induction (*Oyadomari et al., 2002*), significant differences in the number of apoptotic beta-cells were not observed in either model (*Herbach et al., 2007*).

To study the role of proinsulin cysteine disrupting mutations in human beta-cells, we derived human induced pluripotent cell lines (iPSC) from Finnish people carrying C96R (the same cysteine as

the Akita C96Y mutation) and C109Y insulin mutations (*Huopio et al., 2016*) and differentiated them in vitro to beta-like cells. To circumvent the challenges associated with variable iPSC differentiation efficiency, we employed isogenic correction of the cell lines using CRISPR-SpCas9 (*Balboa and Otonkoski, 2015*; *Hsu et al., 2014*). Single-cell RNA sequencing of the in vitro differentiated cells showed increased expression of ER stress-associated transcripts in the INS mutant cells, in concert with reduced proliferation. To further study the properties of these beta-like cells in vivo, we performed transplantation experiments into immunocompromised mice. Transplanted INS mutant beta-like cells exhibited lower insulin secretion and increased levels of ER-stress markers, together with reduced mTORC1 signaling and beta-cell size, without any apparent increase in apoptosis. Our findings suggest that PNDM-associated insulin mutations lead to inadequate development of a functional beta-cell mass.

## Results

### Derivation of induced pluripotent stem cells from people with neonatal diabetes carrying mutations in the insulin gene

We studied two Finnish families with hereditary neonatal diabetes due to heterozygous insulin gene mutations (*Figure 1—figure supplement 1A*). Both missense mutations affect cysteine residues, resulting in the disruption of the proinsulin inter-chain disulphide bonds A7-B7 (mutation C96R) and A20-B19 (mutation C109Y) (*Figure 1B*) which are essential for the proper folding and biological activity of the insulin molecule (*Chang et al., 2003*).

Affected individuals of these families become hyperglycemic 3–4 months after birth. Prior knowledge about the insulin gene mutations in the families enabled the neonatal genetic diagnosis in newborn siblings. Two affected newborns were monitored for the development of the disease. They were born at normal gestational age, presenting normal birth weight and fasting plasma c-peptide levels (*Figure 1—figure supplement 1A*). Continuous subcutaneous glucose monitoring demonstrated the gradual deterioration of glycemia during the first months of life (*Figure 1A*, *Figure 1—figure supplement 1B*).

We derived induced pluripotent stem cells (iPSC) from the affected parents using retroviral and Sendai virus-mediated delivery of reprogramming factors OCT4, SOX2, KLF4 and MYC to dermal fibroblasts obtained from skin biopsies (*Figure 1—figure supplement 2A*). Established iPSC were cultured for at least ten passages and then characterized. They expressed hallmark pluripotency markers, presented normal karyotypes and were able to spontaneously differentiate to the three germ layers in embryoid bodies (*Figure 1—figure supplement 2C–E*). Sequencing of the insulin locus confirmed the presence in heterozygosis of the T to C change causing C96R mutation (iPSC line HEL71.4) (*Figure 1C*) and G to A change causing C109Y mutation (iPSC line HEL107.2) (*Figure 1—figure supplement 2B*).

### CRISPR/Cas9-mediated correction of INS C96R mutation in patient derived iPSCs

Differentiation protocols devised to obtain beta-cells from human pluripotent stem cells (PSC) are not equally efficient across cell lines (*Nostro et al., 2015*; *Sui et al., 2018*). For disease modeling purposes, this variation in the differentiation from iPSC with different genetic backgrounds might obscure the phenotype caused by the mutation under study.

Thus, we generated mutation-free isogenic iPSC lines from the patient-derived HEL71.4 line by correcting the C96R mutation using CRISPR/SpCas9 genome editing (*Figure 1C*). Different guide RNAs were designed to target as close as possible to the C96R mutated codon in the insulin locus and tested in HEK293 cells. A guide RNA (Ins8) cutting 9 base pairs away from the point mutation showed high cutting efficiency by T7 assay (*Figure 1—figure supplement 2F–G*). We tested a mutation correction strategy based on homology directed repair (HDR) stimulated by the Ins8 guide RNA cutting activity using a single stranded DNA oligo (ssODN) of 70 bases as a donor template (*Figure 1C*, *Figure 1—figure supplement 2F*). The correction donor ssODN contained the wild-type nucleotide in the mutation position (wild-type Cys TGT instead of mutant Arg CGT) and a synonymous point mutation in the next codon that created a restriction site for BsrGI (Thr ACC instead of Thr ACA) enabling rapid screening of the recombinant clones. The recombination of the ssODN

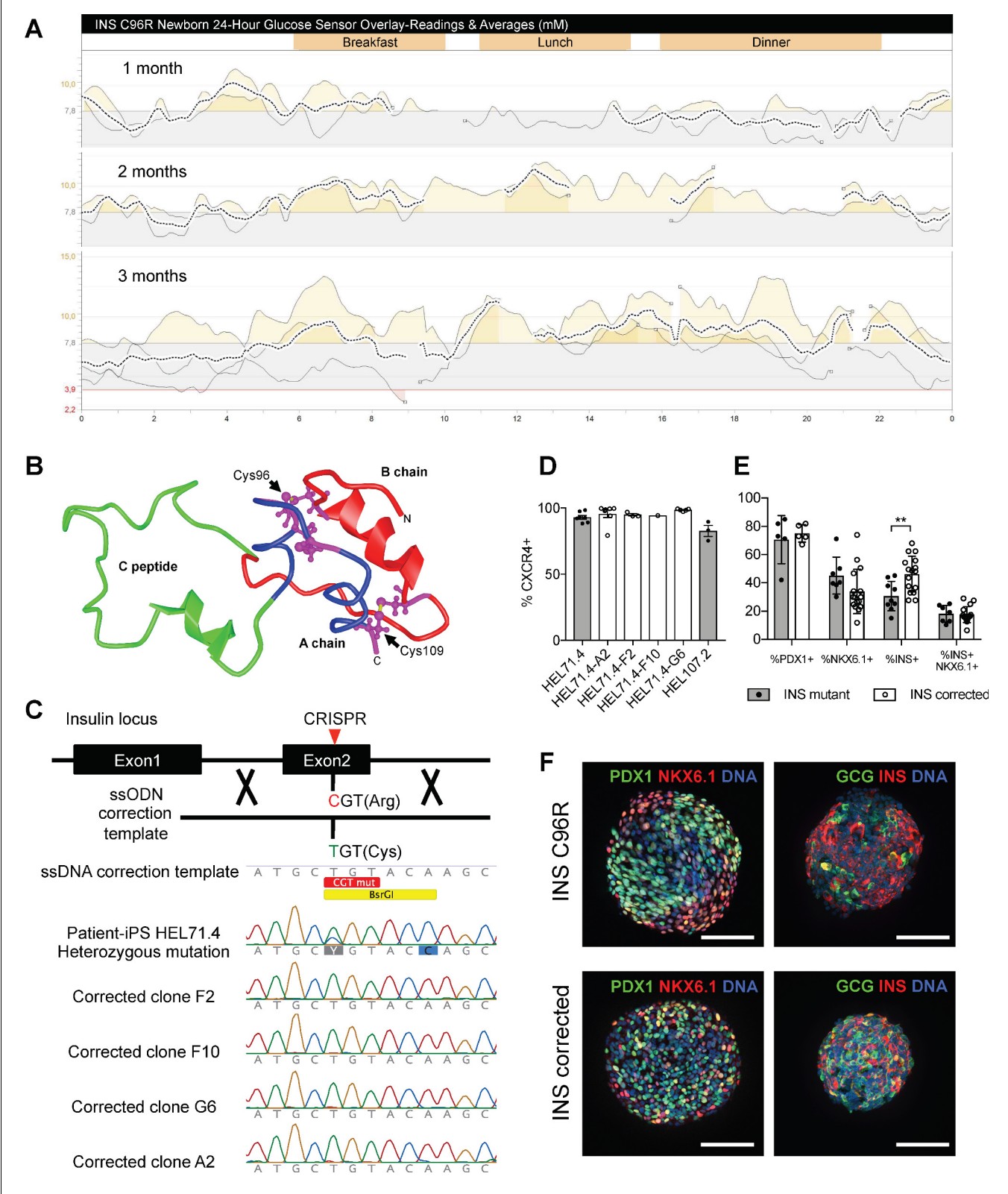

**Figure 1.** Generation of a disease model of neonatal diabetes caused by insulin mutations. (**A**) 24 hr glucose sensor curves of a newborn carrying the INS C96R mutation showing deterioration of glycemic control during the first 3 months of life. (**B**) Proinsulin model depicting the mutated disulfide bridge-forming cysteines. (**C**) Mutation correction strategy mediated by CRISPR/Cas9 stimulated homology directed repair, resulting in four INS C96R mutation corrected iPSC clones. (**D**) Flow cytometry for definitive endoderm marker CXCR4 on day 3 of differentiation of INS mutant and corrected cells

*Figure 1 continued on next page*

*Figure 1 continued*

lines (n = 1–7 independent differentiation experiments per cell line). (E) Flow cytometry for PDX1, NKX6.1 and INS on Stage 7 of differentiation (n = 3–16 independent differentiation experiments). (F) Whole-mount immunostaining for the pancreatic transcription factors PDX1 and NKX2.1 and islet hormones glucagon (GCG) and INS of Stage 7 differentiated islet-like clusters. Scale bars = 100 µm. Data represent mean ± SEM. **p < 0.01, Student's t test.

DOI: https://doi.org/10.7554/eLife.38519.003

The following figure supplements are available for figure 1:

**Figure supplement 1.** Clinical history of the people carrying insulin gene mutations.
DOI: https://doi.org/10.7554/eLife.38519.004
**Figure supplement 2.** iPSC lines characterization and correction of *INS* C96R mutation.
DOI: https://doi.org/10.7554/eLife.38519.005
**Figure supplement 3.** Differentiation efficiency of HEL71.4 mutant and corrected cell lines.
DOI: https://doi.org/10.7554/eLife.38519.006

was efficient in HEK293 cells (*Figure 1—figure supplement 2F–G*). Next, we delivered the SpCas9-T2A-EGFP expressing plasmid, Ins8 gRNA and ssODN to patient iPSC HEL71.4 and sorted them based on EGFP+ expression. Sorted cells were pooled together, expanded and single-cell sorted to 96-well plates. We recovered 27 iPSC colonies out of 384 sorted single cells (Cloning efficiency = 7.03%). We screened the clones by PCR followed by BsrGI restriction and found 17 recombinant clones (17/22 = 77% recombinant) (*Figure 1—figure supplement 2H*). Recombinant clones were further examined by Sanger sequencing (*Figure 1C*). Clones F2, F10, G6 and A2 presented correction of the mutation with no chromosomal abnormalities (*Figure 1—figure supplement 2I*). No mutations in putative off-targets were found in any clone (*Supplementary file 1* – Table 1).

For the differentiation and transplantation experiments in this study we have used the C96R mutant iPSC line HEL71.4 (INS C96R), its corrected clones (INS corrected) and the C109Y mutant iPSC line HEL107.2 (INS C109Y).

## Differentiation of INS mutant and corrected iPSC lines to beta-like cells

We utilized a previously described protocol to differentiate beta-cells from iPSC (*Saarimäki-Vire et al., 2017*) (*Figure 1—figure supplement 3A*). All INS mutant and INS corrected iPSC lines differentiated efficiently into definitive endoderm on day 3 (*Figure 1D*) and progressed to the pancreatic progenitor stage on day 12, presenting abundant PDX1+, NKX6.1+ and SOX9+ cells, with few NEUROG3+ and endocrine CHGA+ cells (*Figure 1—figure supplement 3B*).

At the pancreatic progenitor stage (Stage 4, 12 days of differentiation) cells were dissociated and plated in suspension in a rotational platform, forming 3D islet-like aggregates that differentiated further to the endocrine lineage. After 30 days of differentiation (final Stage 7, S7), we characterized the islet-like aggregates by cytometry. Differentiations from INS mutant and INS corrected iPSC yielded S7 aggregates that were composed of cells expressing PDX1 (an average across all iPSC lines of 73 ± 12% SD PDX1+ cells, n = 10) and NKX6.1 (an average across all iPSC lines of 37 ± 15% SD, NKX6.1+, n = 23) (*Figure 1E*) (*Figure 1—figure supplement 3B–C*). On average across all iPSC lines, 41% (±14% SD, n = 25) of the S7 cells were INS+, with 18% (±6% SD, n = 23) of the cells expressing both INS and NKX6.1, a sign of bona fide beta-cells (*Nostro et al., 2015*) (Results pooled by genotype presented in *Figure 1E–F*, results of individual iPSC lines presented in *Figure 1—figure supplement 3C*). The number of INS+ cells at S7 was significantly different between INS mutant (31 ± 10% SD; n = 9) and corrected cells (49 ± 13% SD; n = 16) (*Figure 1E*, *Figure 1—figure supplement 3C*).

Subsequent analyses focused particularly on the INS+ cells at the Stage 7 of in vitro differentiation or 1, 3 or 6 months after transplantation under the kidney capsule of immunodeficient NSG mice.

## Single-cell RNA sequencing revealed increased ER-stress and impaired proliferation of INS mutant beta-like cells

Bulk RNA isolation has been traditionally used to study gene expression in cell samples using RT-qPCR or RNA-seq analysis. Using these methods, the identity of the cell source of a given RNA transcript is lost. Moreover, cell sample heterogeneity might introduce a lot of variation when studying a

particular cell-specific transcript. Of note, independent iPSC to beta-like cell differentiation experiments vary in the yield of INS+ cells (*Figure 1—figure supplement 3C*). The use of bulk RNA analysis to study the effects of the INS mutation in the beta-like cells might therefore obscure subtle transcriptional differences between the INS mutant and corrected cells. To overcome this problem we performed InDrop droplet-based single-cell RNA sequencing (scRNAseq) on differentiated islet-like cells (*Baron et al., 2016*; *Klein et al., 2015*).

A total of 2 287 single cells from INS C96R and INS corrected Stage 7 islet-like aggregates were sequenced with a mean depth of 38 329 aligned reads/cell. Out of these cells, 2 171 (94.9%) passed quality control. An average of 3 321 unique transcripts (UMI) and 1391 genes were detected per cell (*Figure 2—figure supplement 1A*, *Supplementary file 1* – Table 9, *Figure 2—source data 1*). Clustering analysis of all cells from both genotypes distinguished four different cell populations, expressing markers of beta-cells (beta-like cells), endocrine progenitor cells (progenitor cells), alpha cells (alpha-like cells), and proliferating alpha cells (proliferating alpha-like cells) (*Figure 2A–B*, *Figure 2—figure supplement 1*, *Figure 2—figure supplement 4A–B*) (*Supplementary file 1* – Table 2) (*Segerstolpe et al., 2016*). The highest levels of *INS* expression were observed in the beta-like cluster (*Figure 2B–C*). A resampling procedure confirmed the robustness of the clustering results (*Figure 2—figure supplement 4C–D*).

To confirm and to strengthen the identity of the sequenced cells, we compared our scRNA-seq data with a previous published human adult islet scRNAseq dataset generated with the InDrop single-cell platform (*Baron et al., 2016*). Mapping of the individual cells to the *Baron et al. (2016)* dataset confirmed the identity of the beta-cells and was utilized to further refine the clustering of the beta-like and progenitor clusters, filtering out cells that mapped to the alpha cell cluster (*Figure 2—figure supplement 1B*, *Figure 2—figure supplement 4A–B*).

We performed differential gene expression analysis on both refined beta-like and progenitor clusters between INS C96R and INS corrected cells to determine the transcriptional changes caused by the INS C96R mutation at the single-cell level (*Figure 2D*, *Figure 2—figure supplement 1C*) (*Supplementary file 1* – Tables 3 and 4). Mutant beta-like cells presented significant upregulation of chaperone genes *HSPA5*, *HSPA8* and *HSP90B1*, disulfide isomerase *PDIA6*, ubiquitin gene *UBB* and components of the ER associated degradation (ERAD) pathway *HM13*, *HERPUD1*, *SEC61B* and *SDF2L1*. SDF2L1 expression is induced upon overexpression of the Akita mouse model C96Y mutant INS and it has been shown to interact with misfolded proinsulin and mediate its degradation (*Figure 2—figure supplement 1C*) (*Hartley et al., 2010*; *Tiwari et al., 2013*). These transcriptional changes indicate that the INS C96R mutation causes ER-stress and the subsequent activation of the UPR. Expression of the *CDKN1C* gene was also increased. The gene product, p57/Kip2, is an important inhibitor of human beta-cell proliferation (*Avrahami et al., 2014*), suggesting reduced proliferation of the INS C96R cells. Interestingly, the *INS* gene was upregulated in INS C96R cells, together with other insulin secretion related genes (*CPE*, *SCGN*, *DLK1*), a phenomenon previously described in young Akita mice (*Oyadomari et al., 2002*) (*Figure 2D*, *Figure 2—figure supplement 1C*, *Supplementary file 1* – Tables 3 and 4).

Conversely, genes encoding mitochondrial respiratory chain subunits (*MT-CO1*, *MT-CO2*), immediate early gene *IER2*, transcription factors *PAX6* and *RFX6*, and the mTOR regulator *LAMTOR5* were significantly downregulated in INS C96R beta-like cells (*Figure 2D*, *Figure 2—figure supplement 1C*). Similar to the beta-like cluster cells, INS C96R progenitor cells presented increased *INS* transcription and downregulated immediate early genes (*FOS*, *EGR1*, *IER2*, *JUN*), mitochondrial genes (*MT-CO1*, *MT-CO2*, *MT-CYB*), pancreatic transcription factors (*FOXA2*, *GATA6*, *RFX6*, *PAX6*) and progenitor proliferation-associated ID genes (*ID2*, *ID3*) (*Figure 2D*, *Figure 2—figure supplement 1C*). These results indicate that the ER-stress caused by the INS C96R proinsulin results in downregulation of genes associated with beta-cell proliferation and function.

Gene set enrichment analysis (GSEA) was performed on the differentially expressed genes in the beta-cell cluster. Ribosomal translation, insulin synthesis and processing, diabetes pathways and ATF6-controlled UPR chaperones gene sets were overrepresented among the genes upregulated in INS C96R cells. On the contrary, the genes downregulated in INS C96R were enriched in gene sets related to EGF signaling and respiratory electron transport, processes important for beta-cell development and function. In the progenitor cluster, GSEA showed overrepresentation of GLIS3 targets and peptide chain elongation in the INS C96R upregulated genes, while SRC, TGF-beta and EGF

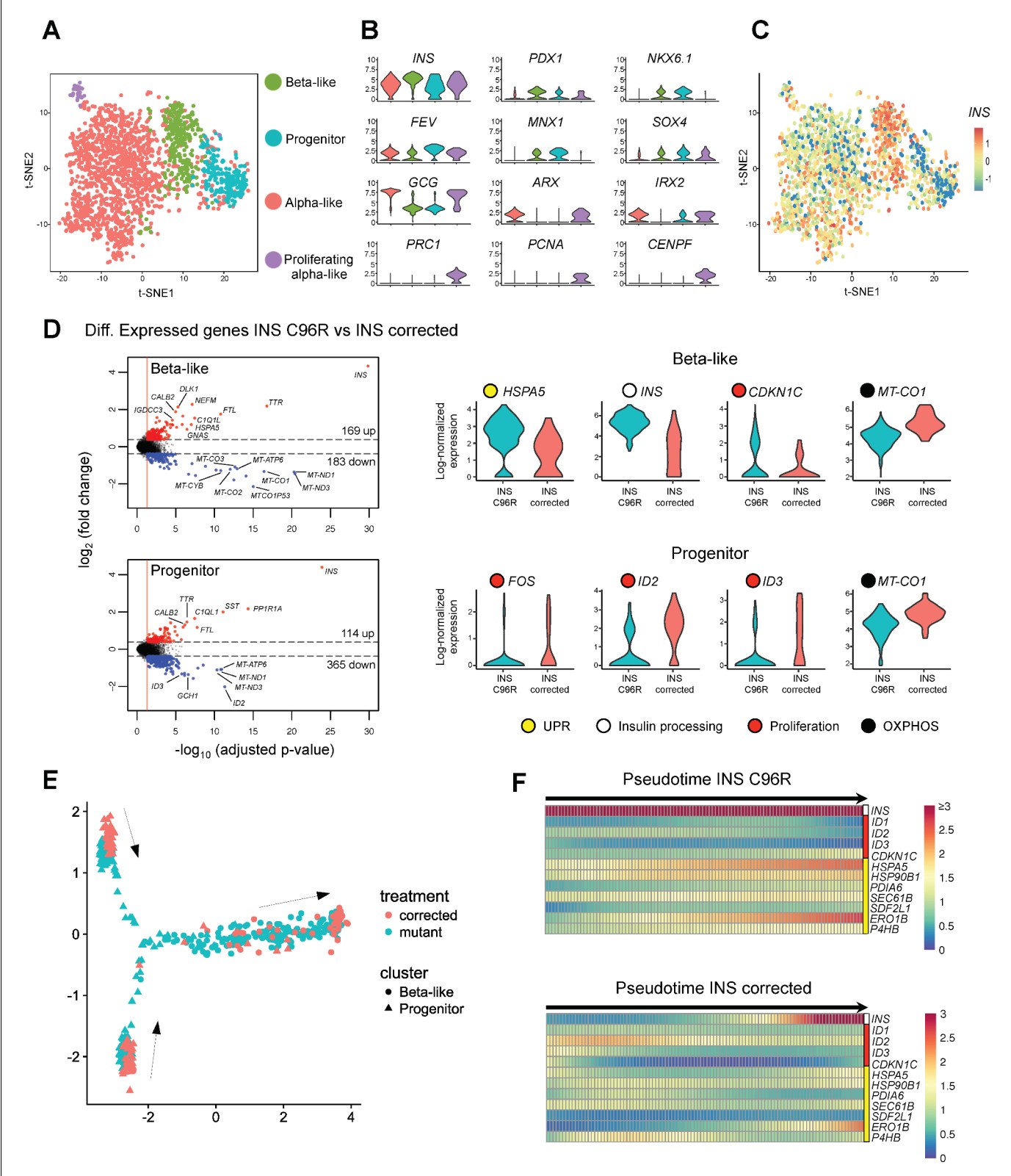

**Figure 2.** Single cell RNA sequencing revealed increased ER-stress and reduced proliferation in INS mutant beta-like cells. (**A**) Single cell RNAseq clustering analysis of Stage 7 islet-like aggregates cells derived from both INS C96R and INS corrected iPSC. A total of 1991 post-QC cells mapped to the *Baron et al. (2016)* dataset were used for clustering. Four distinct clusters were identified: beta-like cells (308 cells), endocrine progenitor cells (236 cells), alpha-like cells (1252 cells) and proliferating alpha-like cells (45 cells). (**B**) Violin plots showing log-normalized expression of selected marker genes

*Figure 2 continued on next page*

*Figure 2 continued*

for each cluster. See also *Supplementary file 1* – Table 2. (C) Expression of *INS* in the different cell populations clusters. (D) Volcano plots illustrate the differentially expressed genes between INS C96R and INS corrected cells in beta-like and progenitor clusters. Violin plots show the relative expression of *INS*, unfolded protein response (UPR) gene *HSPA5 (BIP)*, proliferation and oxidative phosphorylation related genes that are differentially expressed (fold change ≥ 1.3, adjusted p value < 0.05) between INS C96R and INS corrected cells in the beta-like and progenitor clusters. See also *Supplementary file 1* – Table 3. (E) Differentiation trajectory inferred from pseudotime analysis of the beta-like and progenitor clusters. (F) Heatmaps show the normalized, smoothed expression of *INS*, proliferation (red) and ER-stress (yellow) genes that are differentially regulated across pseudotime between INS C96R and INS corrected cells.

DOI: https://doi.org/10.7554/eLife.38519.007

The following source data and figure supplements are available for figure 2:

**Source data 1.** Single-cell RNA-seq gene count matrices and alignment statistics.

DOI: https://doi.org/10.7554/eLife.38519.012

**Source data 2.** HALLMARK_APOPTOSIS and curated C2 gene sets from Broad Institute's Molecular Signatures Database (MSigDB).

DOI: https://doi.org/10.7554/eLife.38519.014

**Figure supplement 1.** Single cell RNA sequencing data analysis strategy.

DOI: https://doi.org/10.7554/eLife.38519.008

**Figure supplement 2.** Pseudotime analysis of INS C96R vs INS corrected cells.

DOI: https://doi.org/10.7554/eLife.38519.009

**Figure supplement 3.** Single-cell RNA-seq quality control with mitochondrial and apoptosis count filters.

DOI: https://doi.org/10.7554/eLife.38519.010

**Figure supplement 4.** Cluster composition and robustness.

DOI: https://doi.org/10.7554/eLife.38519.011

signaling gene sets were enriched in the downregulated genes (*Supplementary file 1* – Table 5, *Figure 2—source data 2*).

Mitochondrial oxidative phosphorylation, regulation of macromolecule metabolic processes and mitotic cell cycle Gene Onthology (GO) biological processes were enriched among the genes downregulated in the beta-like cluster INS C96R cells. Intracellular transport, protein targeting to ER and protein folding GO terms were enriched among the upregulated genes. Similar GO terms were enriched in the down and upregulated genes of the progenitor cluster (*Supplementary file 1* – Table 6, *Figure 2—source data 2*).

We hypothesized that the endocrine progenitor population identified by the clustering is in the process of differentiating towards endocrine (mainly alpha-like and beta-like) cells. To investigate the hierarchy of differentiation events, we performed pseudotime analysis to infer the possible trajectories of these populations (*Qiu et al., 2017*). The analysis algorithm imposed a branched trajectory, in which the progenitor cluster cells were split in two distinct progenitor branches that merge to give rise to the beta-like cell branch (*Figure 2E*, *Figure 2—figure supplement 2A*). The differentially expressed genes between these two progenitor populations suggest that Progenitor one may represent early endocrine progenitors, expressing higher levels of *HES1*, *PAX6*, *PROX1* and other genes related to cytoskeleton regulation, cell adhesion, migration and TGF-beta signaling modulation (*Figure 2E*, *Figure 2—figure supplement 2A–C*, *Supplementary file 1* – Table 7). The Progenitor two population presents higher levels of *NEUROD1*, *CHGA* and *FEV*, suggesting a more advanced, already granulated, late endocrine progenitor identity.

Analysis of differentially regulated genes along the pseudotime trajectory showed that the expression dynamics of *INS* and other proliferation (*ID2*, *CDKN1C*) and UPR-related genes (*HSPA5*, *PDIA6*, *SDF2L1*) were significantly different between INS C96R and INS corrected cells (*Figure 2F*, *Figure 2—figure supplement 2D*, *Supplementary file 1* – Table 8). This illustrates the significantly higher upregulation of ER-stress markers, *INS* and *CDKN1C* cell cycle inhibitor earlier in the differentiation trajectory of INS C96R progenitors to beta-like cells, indicating the immediate negative effect of misfolded proinsulin in the recently committed beta-like cells upon *INS* expression.

## Increased ER-stress and reduced proliferation of INS mutant beta-like cells in vitro

To determine the expression of different ER-stress markers at the protein level, we performed immunohistochemistry of stem cell-derived Stage 7 beta-like cells cultured in vitro. Immunoreactivity for

ER-stress associated proteins BIP (HSPA5), GRP170 (HYOU1) and MANF was significantly increased in the INS C96R beta-like cells as compared with the corrected cells (*Figure 3A–B*, *Figure 5—figure supplement 1C*). Conversely, the number of KI67+ and PCNA+ proliferative insulin-positive cells was significantly reduced in the INS C96R cells (*Figures 3A–B and* 6E–F). We quantified single cell levels of INS immunostaining intensity using flow cytometry and from immunohistochemistry preparations (*Figure 1—figure supplement 3D–E*). We did not find significant differences in the signal intensity between INS mutant and INS corrected cells, indicating that impaired proliferation of INS mutant beta-like cells is likely the major contributor to the reduced percentage of INS+ cells observed by cytometry.

qRT-PCR analysis confirmed the significantly increased expression of ER-stress markers BIP, sXBP1, MANF and GRP170, in line with the findings of immunostainings and scRNAseq results (*Figure 3C*). Other ER-stress associated genes, including *CHOP (DDIT3)*, which has been reported to be upregulated in Akita mice (*Oyadomari et al., 2002*), were not differentially expressed at the mRNA level between INS C96R and corrected cells (*Figure 3C*). However, the expression of CHOP and ATF transcription factors might be also regulated post-transcriptionally (*Cnop et al., 2017*). To determine if the INS C96R beta-like cells under ER-stress are more sensitive to apoptosis, TUNEL assays were performed on S7 aggregates. The ratio of TUNEL+/INS+ cells was similar in INS C96R and INS corrected cells in the basal conditions (about 1%) (*Figure 3D*). Induction of additional ER-stress by treatment with brefeldin A (BFA), thapsigargin (TGA) or tunicamycin (TM) resulted in increased apoptosis for both genotypes. BFA, but not the other stressors, induced a significantly higher level of apoptosis in the INS C96R cells compared to mutation corrected cells (*Figure 3D*). Thus, INS C96R beta-like cells present higher sensitivity to apoptotic cell death induced by further increasing ER-stress with BFA treatment.

The insulin secretory responses of S7 beta-like cells were assessed by sequential static incubations in the presence of 1 µM forskolin to increase cAMP levels. While the response to high glucose alone was minimal (1.15- and 1.05-fold for INS C96R and corrected cells, respectively), tolbutamide and KCl triggered robust insulin secretion (2.5- and 3.2-fold for both INS C96R and corrected cells). The fractional release of insulin (as % of content) was not significantly different between INS C96R and corrected cells (*Figure 3E*). However, the insulin content of the INS C96R S7 islet-like aggregates was significantly reduced (5.2-fold lower than in corrected) (*Figure 3F*). The ratio of human proinsulin content to human insulin content was not significantly different between INS C96R and corrected cells at this stage (*Figure 3—figure supplement 1A*), but the total proinsulin content was significantly reduced in INS C96R (*Figure 3—figure supplement 1B*) as well as the ratio of secreted proinsulin in maximal stimulation with KCl (*Figure 3—figure supplement 1C*). Taken together, these results show that a) Stage 7 beta-like cells are not functionally mature enough to respond to glucose stimulation alone, and b) the INS C96R beta-like cells maintain their responsiveness to pharmacological stimulation despite markedly decreased proinsulin and insulin content.

## INS mutant beta-like cells presented reduced insulin secretion after in vivo transplantation

Islet-like cell clusters from INS C96R, INS C109Y and corrected cell lines were transplanted under the kidney capsule of NSG mice to study the effect of the insulin mutation on beta-like cells in vivo (*Figure 4A*). Graft functionality was tracked from 1 to 6 months after transplantation by measuring circulating human C-peptide in plasma samples from randomly fed mice. Mice carrying INS C96R and INS C109Y grafts presented significantly lower levels of human C-peptide than mice carrying INS corrected grafts (*Figure 4B*). The levels of C-peptide increased from the 2 month time point onwards for the INS corrected grafts (p < 0.0001, One-way ANOVA), while no change was observed in the INS C96R and INS C109Y grafts (*Figure 4B*). The increase in circulating human C-peptide in INS corrected grafts can be attributed to further expansion and maturation of the transplanted beta-like cells. Additionally, differentiation of co-transplanted pancreatic progenitors to the beta-cell lineage is likely to contribute as well.

Intraperitoneal glucose tolerance test performed at 3 or 6 months after grafting showed that INS corrected grafts presented higher insulin secretion levels than INS C96R and INS C109Y grafts, both upon fasting and after glucose injection (*Figure 4C–D*). Mice transplanted with INS C96R grafts presented a trend towards elevated ratio of circulating proinsulin to circulating C-peptide (*Figure 4E*), a

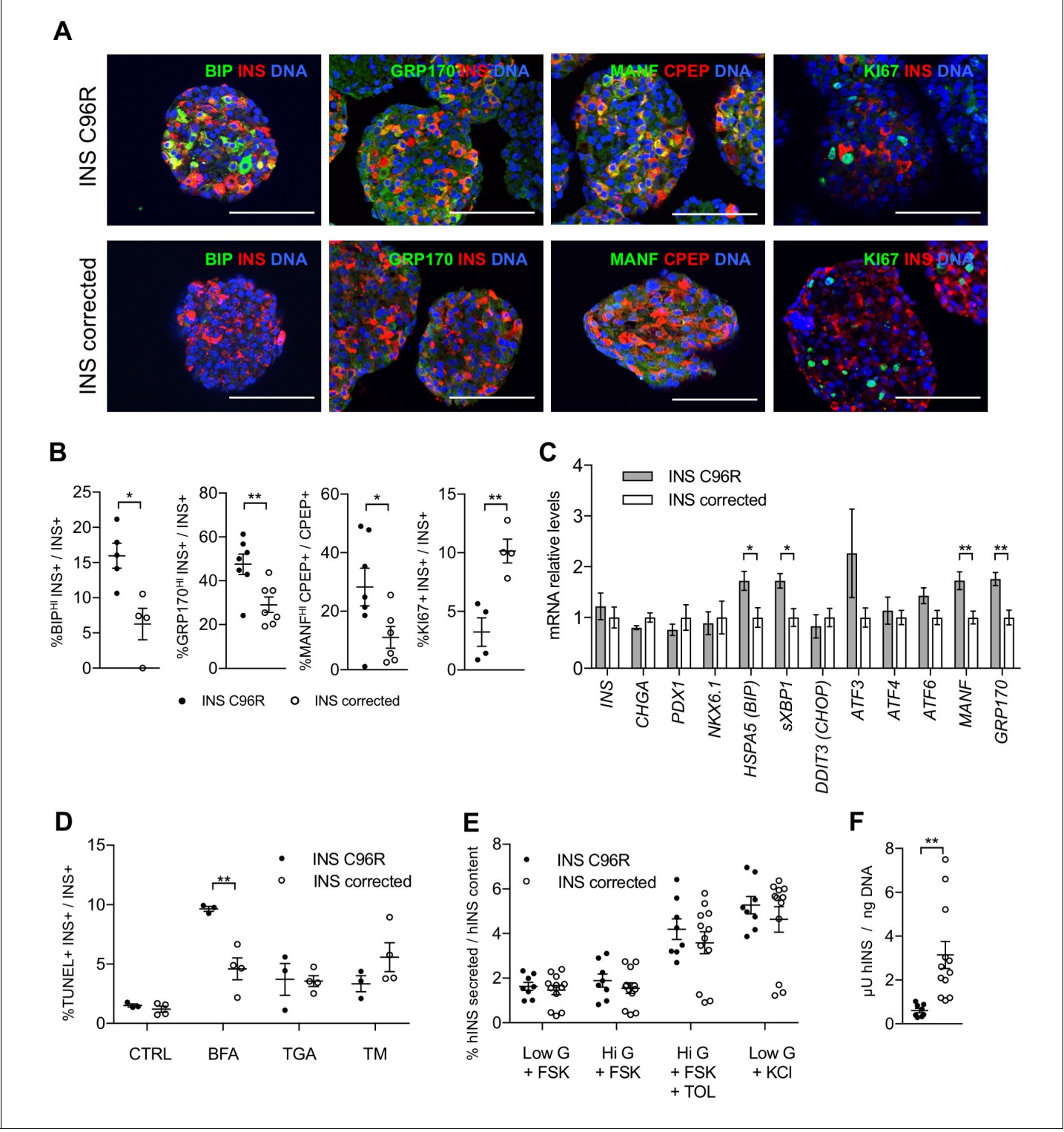

**Figure 3.** In vitro differentiated INS mutant cells presented increased ER-stress associated with reduced proliferation and insulin content. (A) Immunohistochemistry for ER-stress markers (BIP, GRP170, MANF) and proliferation marker KI67 along with INS in Stage 7 cells. Scale bars = 100 μm. (B) Quantification of (A). Percentage of Stage 7 insulin positive cells that express BIP, GRP170, MANF and KI67 (n = 3–7 independent differentiation experiments per genotype). (C) qRT-PCR of beta-cell and ER-stress markers (n = 5–6 independent differentiation experiments per genotype). (D) Sensitivity to ER-stress-induced apoptosis of Stage 7 cells. Percentage of insulin positive cells that are labeled by TUNEL assay in control conditions and after treatment with ER-stress inducers brefeldin A (BFA), thapsigargin (TGA) and tunicamycin (TM). (E) Static sequentially stimulated insulin secretion of
*Figure 3 continued on next page*

*Figure 3 continued*

Stage 7 cells, presented as fractional secretion of total INS content. Low G = 3.3 mM glucose; Hi G = 20 mM glucose; TOL = tolbutamide 100 µM; KCl = 30 mM KCl; FSK = 1 µM forskolin. (n = 8–12 independent stimulations per genotype). (F) Human insulin content in Stage 7 cells. Cell mass normalized by DNA content (n = 8–12 independent stimulations per cell genotype). Data represent mean ± SEM. Student's t test, *p < 0.05, **p < 0.01.
DOI: https://doi.org/10.7554/eLife.38519.015

The following figure supplement is available for figure 3:

**Figure supplement 1.** Characterization of in vitro Stage 7 cells and 3 months old grafts.
DOI: https://doi.org/10.7554/eLife.38519.016

phenomenon that has been previously described in humans with mutant insulin diabetes (*Liu et al., 2010b*; *Liu et al., 2015*; *Rajan et al., 2010*).

## Elevated ER-stress levels in INS mutant beta-like cells after transplantation without increased apoptosis

Grafts were retrieved at 1, 3 and 6 months and examined by immunohistochemistry. Almost all INS C96R and INS C109Y INS+ cells in 3-month-old grafts presented higher levels of immunoreactivity for proinsulin in comparison to INS corrected grafts (*Figure 4F–H*). A similar pattern was observed in the S7 cells, as well as 1 month and 6 month grafts (*Figure 5—figure supplement 1D*). Proinsulin immunoreactivity occupied most of the cytoplasm in INS C96R cells, while it was punctate in INS corrected cells. This indicates defective proinsulin transport, resulting in its accumulation (*Figure 4G*).

Immunohistochemistry for ER-stress markers revealed profound differences in the expression of BIP, MANF and GRP170 at 1, 3 and 6 months, with remarkably increased levels in the INS C96R and C109Y cells compared with INS corrected cells (*Figure 5A–E* and *Figure 3—figure supplement 1E* for 3 months grafts) (*Figure 5—figure supplement 1* for 1 and 6 month grafts, *Figure 5—figure supplement 1C* shows the quantification of BIP intensity on a per cell basis). The percentage of INS C96R beta-like cells expressing high levels of BIP increased dramatically after transplantation (*Figure 5F*, *Figure 5—figure supplement 1C*), indicating a progressive aggravation of ER-stress in vivo. Mild UPR increase has been shown to induce beta-cell proliferation in hyperglycemic conditions (*Sharma et al., 2015*). We examined the relationship between the proliferation marker PCNA and the ER-stress marker BIP, but we did not find a consistent correlation (*Figure 5—figure supplement 2A*). We performed a similar analysis for proliferation marker KI67 and the intensity of INS and PROINS immunostaining, since decreased levels of INS expression have been associated with increased beta-cell proliferation (*Szabat et al., 2016*; *Xin et al., 2018*). KI67+ cells presented reduced INS immunostaining intensity across genotypes and time points, suggesting a possible correlation between reduced INS expression and increased proliferation (*Figure 5—figure supplement 2B–C*).

Similar to proinsulin, MANF immunoreactivity occupied most of the cytoplasm in INS C96R cells, indicating accumulation in a distended ER (*Figure 5D*)(*Lindahl et al., 2014*). To detect if MANF was released from the cells under ER-stress, we measured it in vitro and in transplanted animals. Overnight MANF secretion tended to be elevated in the INS C96R in vitro cells, but this difference was not significant (*Figure 3—figure supplement 1D*). Circulating human MANF levels were below detection limits in the blood of transplanted animals.

Apoptotic INS+ cells assayed by TUNEL or CASP3 staining were very rare at all time points (<0.5% at 3 months, not significantly different) (*Figure 3—figure supplement 1F*), suggesting that aggravated ER-stress does not lead to increased apoptosis of the INS C96R or INS C109Y cells in vivo.

## INS mutant grafts presented altered endocrine cell proportions and PDX1 expression

The proportion of INS+ cells was significantly reduced in INS C96R grafts at 3 months. This finding was further confirmed by immunohistochemistry for C-peptide (CPEP) (*Figure 6A–B*). On the contrary, the glucagon (GCG) positive cell compartment was increased (*Figure 6A–B*). We examined other endocrine hormones and found that there were no significant differences in the ratios of somatostatin (SST) positive or pancreatic polypeptide (PP) positive cells. However, the percentage

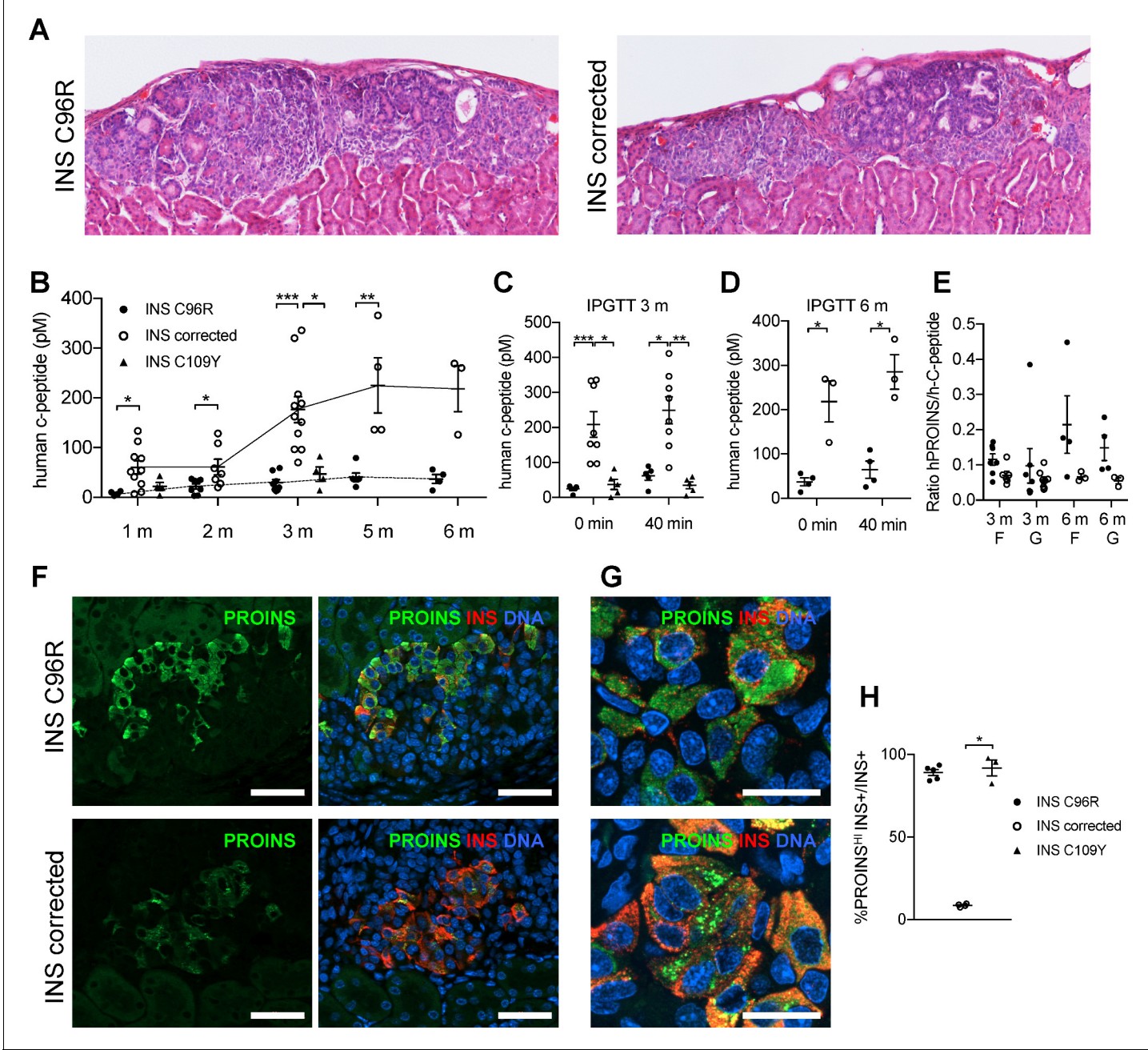

**Figure 4.** Reduced insulin secretion and increased proinsulin accumulation in transplanted INS mutant beta-cells. (A) Hematoxylin-eosin staining of Stage 7 islet-like cell clusters transplanted under the kidney capsule of NSG mice and retrieved after 3 months. (B) Monthly tracking of INS C96R, INS C109Y and INS corrected grafts functionality by measuring circulating human C-peptide in randomly fed transplanted mice (n = 3–11 independent transplanted animals per cell genotype and time point; Kruskal-Wallis test for 1 and 3 months, Mann-Whitney test for 2, 5 and 6 months). (C–D) Intraperitoneal glucose tolerance test (IPGTT) in mice transplanted for 3 and 6 months. C-peptide levels measured on fasted animals 0 min and 40 min after glucose injection. (n = 5–8 independent transplanted animals per cell genotype and time points; Kruskal-Wallis test for (C), Mann-Whitney test for (D)). (E) Ratio of human proinsulin to human C-peptide in fasted mice (F) and 40 min after glucose injection (G) at 3 and 6 months after transplantation (n = 3–8 independent transplanted animals per cell genotype and time point). (F) Immunohistochemistry for insulin (INS) and proinsulin (PROINS) in 3-months old grafts. Scale bars = 50 μm. (G) Higher magnification of immunohistochemistry for insulin (INS) and proinsulin (PROINS) in 3-months old grafts. Scale bars = 20 μm. (H) Percentage of INS+ cells stained for PROINS in 3 month old grafted beta-like cells (n = 3–5; Kruskal-Wallis test). Data represent mean ± SEM. *p < 0.05, **p < 0.01, ***p < 0.001.
DOI: https://doi.org/10.7554/eLife.38519.017

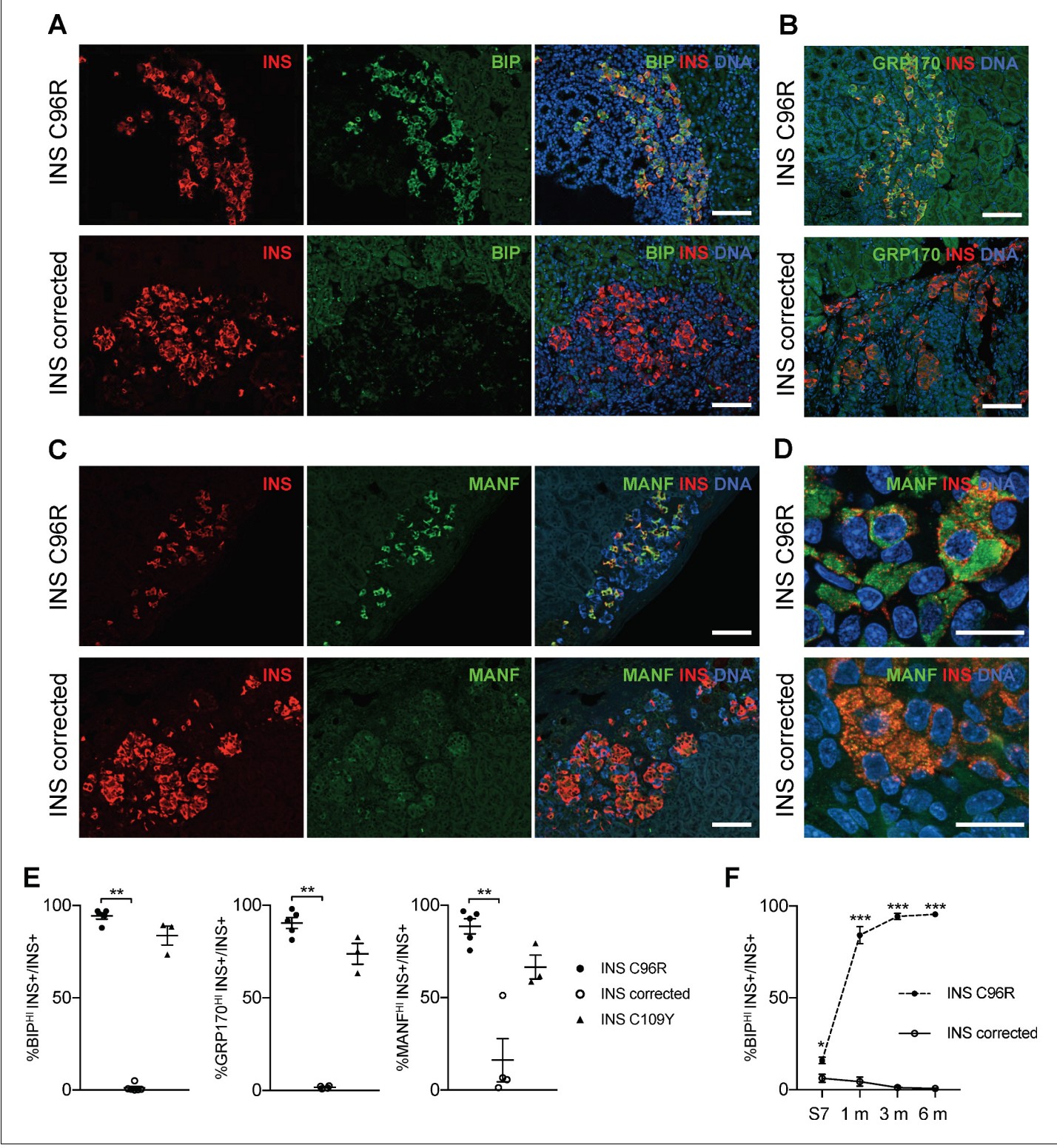

**Figure 5.** Transplanted INS mutant beta-cells presented increased expression of ER-stress markers. (A–C) Immunohistochemistry for ER-stress markers BIP, GRP170 and MANF together with INS in 3 months old grafts. Scale bars = 100 μm. (D) Closer magnification of immunohistochemistry for MANF. Scale bars 20 μm. (E) Quantification of (A–C). Percentage of insulin positive cells expressing BIP, GRP170 or MANF in 3 months old grafts (n = 3–5 independent transplanted animals per genotype; Kruskal-Wallis test). (F) Dynamic changes in the percentage of insulin positive cells expressing BIP between Stage 7 and 6 months old grafts (n = 3–6; Student's t test). Data represent mean ± SEM. *p < 0.05, **p < 0.01, ***p < 0.001.

*Figure 5 continued on next page*

*Figure 5 continued*

DOI: https://doi.org/10.7554/eLife.38519.018

The following figure supplements are available for figure 5:

**Figure supplement 1.** Increased ER-stress and reduced proliferation in 1 and 6–month grafts.

DOI: https://doi.org/10.7554/eLife.38519.019

**Figure supplement 2.** Correlation of proliferation markers with ER-stress marker levels in INS+ cells.

DOI: https://doi.org/10.7554/eLife.38519.020

of ghrelin (GHRL) positive cells was significantly increased in INS C96R grafts (*Figure 6A–B*). Double hormone positive INS+GCG+, CPEP+GCG+, INS+SST+, INS+GHRL+ or CPEP+PP+ cells were very rare (<2%) and not significantly different between INS C96R and corrected grafts (*Figure 6C*).

The proportion of cells co-expressing PDX1 and C-peptide was significantly reduced in 3-month-old grafted INS C96R beta-cells (*Figure 6A,D*). Proliferation of INS C96R beta-like cells was significantly impaired at S7 in vitro (*Figures 3B* and *6E–F*). Following transplantation, this difference gradually disappeared as the proliferation of INS corrected cells decreased (*Figure 6D–F*). This may recapitulate human postnatal beta-cell development, with a postnatal peak of proliferation that declines rapidly during the first two years of life (*Gregg et al., 2012*; *Meier et al., 2008*).

## mTORC1 signaling was dysregulated in INS+ mutant cells

mTORC1 signaling is required for the proper postnatal growth and maturation of beta-cells (*Ni et al., 2017*; *Sinagoga et al., 2017*). Single-cell RNA-seq revealed downregulation of genes involved in proliferation, oxidative phosphorylation and mTORC1 regulation (*LAMTOR5*) in INS C96R beta-like cells. These processes are in part regulated by mTORC1 signaling, suggesting that it could be dysregulated in INS C96R cells. We found that immunoreactivity for phosphorylated S6, a central downstream signaling target of mTORC1, was significantly reduced in INS+ 3 month-old INS C96R grafts (*Figure 7A–C*). Consistent with reduced mTORC1 signaling, INS+ cell size was also significantly reduced in the INS C96R grafts (*Figure 7B–C*).

Based on the findings of the sc-RNAseq analysis on the in vitro beta-like cells, we examined the expression of mitochondria respiratory chain subunits in 3 month old grafts (*Figures 2D* and *7D*). Immunostaining for MT-CO1, MT-CO2 and TOM20 revealed an altered, more condensed and globular mitochondrial morphology as well as decreased immunofluorescence intensity in INS + mutant cells (*Figure 7D–E*). Interestingly, INS staining intensity was also decreased at this point, suggesting a reduced INS protein content in the grafted beta-like cells (*Figure 7E*).

## Discussion

We interrogated the impact of insulin gene mutations on beta-cell development by generating a model based on genome edited patient-derived iPSC (*Balboa and Otonkoski, 2015*; *Saarimäki-Vire et al., 2017*; *Shang et al., 2014*; *Zhu et al., 2016*). iPSC provide a novel possibility to study mechanisms of beta-cell dysfunction using patient cells. However, this approach is still being developed and presents important caveats that need to be taken into account. The in vitro-differentiated beta-like cells are functionally immature, which limits their usefulness particularly to model metabolically controlled insulin secretion. Also, the variability of differentiation efficiency across cell lines has been a confounding factor. Genome editing technologies have partially solved this problem by enabling the generation of isogenic cell lines. We combined this approach with single-cell transcriptomics, enabling the detection of ER-stress early during the development of beta-cells as the consequence of accumulation of misfolded mutant proinsulin. Impaired beta-cell proliferation was the most striking phenotype of the mutated cells. Following transplantation, the INS mutant cells presented increased proinsulin accumulation and further increased signs of ER-stress, associated with reduced PDX1 expression and reduced beta-cell size, as well as mitochondrial alterations. Many of these features are attributable to the observed decreased mTORC1 signaling.

Since populations of endocrine cells generated with the available differentiation protocols (*Pagliuca et al., 2014*; *Rezania et al., 2014*; *Russ et al., 2015*) are heterogeneous, single-cell RNA-seq methods provide a robust approach to identify and characterize specifically bona-fide beta-like

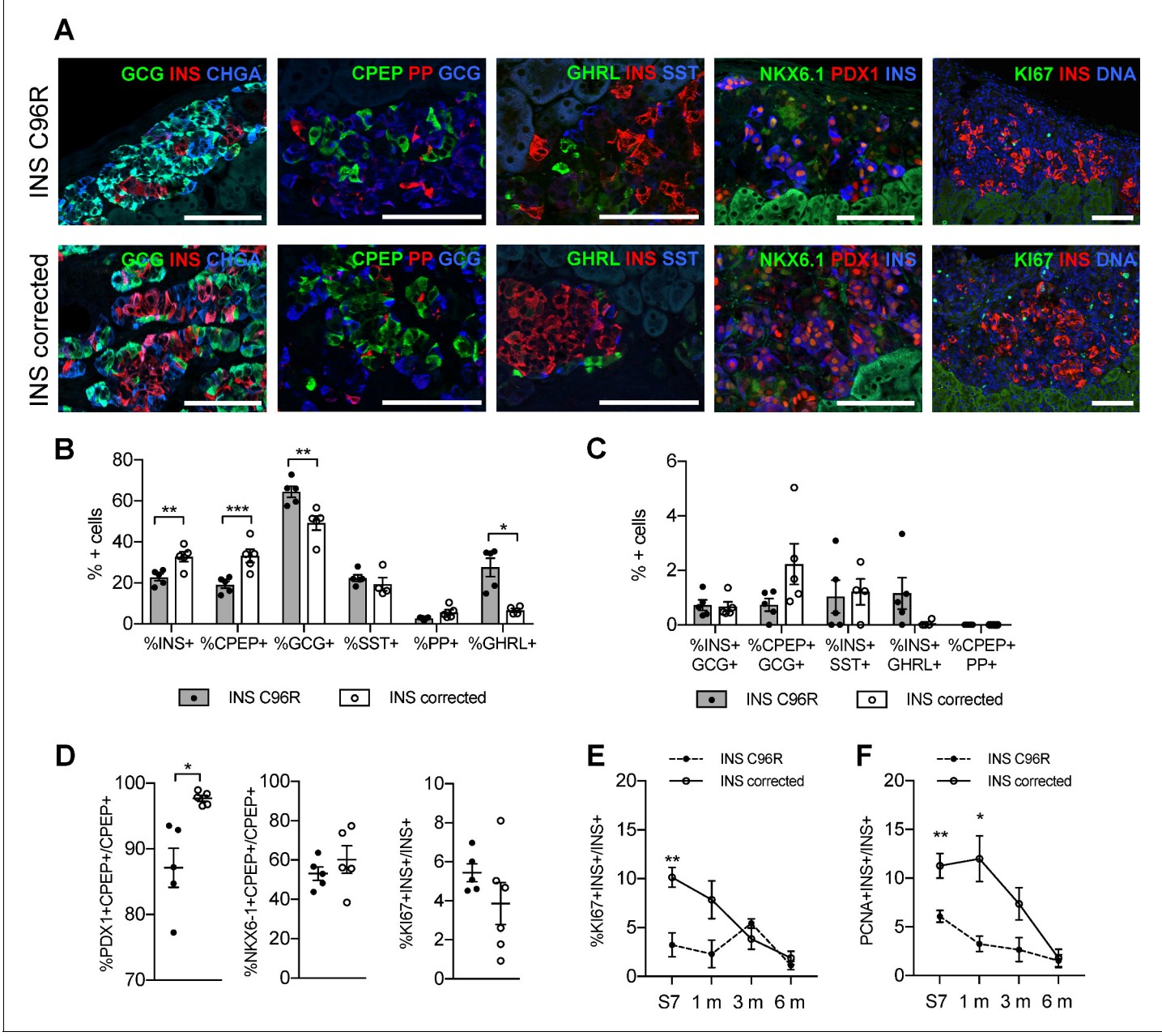

**Figure 6.** Altered endocrine cell proportions and reduced PDX1 expression in INS mutant grafts. (**A**) Immunohistochemistry for endocrine hormones glucagon (GCG), insulin (INS), chromogranin A (CHGA), C-peptide (CPEP), pancreatic polypeptide (PP), ghrelin (GHRL), somatostatin (SST), transcription factors PDX1 and NKX6.1, and proliferation marker MK67 (KI67) on 3 months old grafts. Scale bars = 100 μm. (**B**) Quantification of immunohistochemistry presented in (**A**) for proportions of monohormonal cells in 3 months old grafts. (n = 4–5, Student's t test for all except for GHRL where Mann-Whitney test was used) (**C**) Proportions of polyhormonal cells in 3 months old grafts. (**D**) Quantification of immunohistochemistry presented in (**A**). Percentage of c-peptide/insulin positive cells expressing PDX1, NKX6.1 or KI67 in 3 months old grafts (n = 5–6 independent transplanted animals per genotype, Student's t test with Welch's correction). (**E**) Dynamic changes in the percentage of insulin positive cells expressing KI67 between Stage 7 and 6 months old grafts (n = 4–6; Student's t test). (**F**) Dynamic changes in the percentage of insulin positive cells expressing PCNA between Stage 7 and 6 months old grafts (n = 4–6; Student's t test). Data represents individual values and mean ± SEM. See also *Figure 5—figure supplement 1*.
DOI: https://doi.org/10.7554/eLife.38519.021

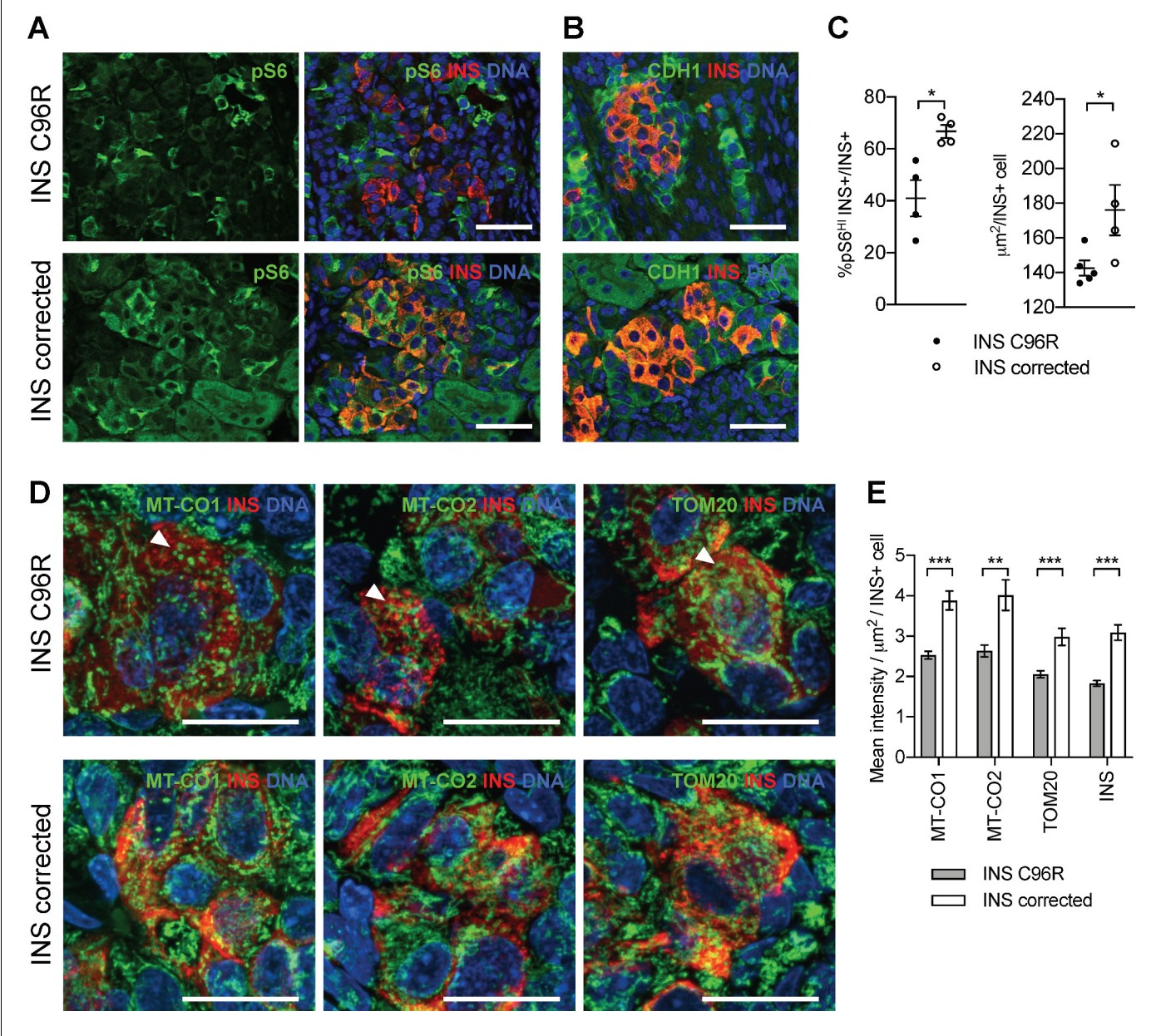

**Figure 7.** Transplanted INS mutant beta-like cells presented reduced mTORC1 signaling, reduced size and decreased mitochondrial respiratory chain subunits expression. (**A**) Immunohistochemistry for the mTORC1 activity marker pS6 and INS in 3 months old grafts. (**B**) Immunohistochemistry for E-Cadherin (CDH1) and INS to quantify beta-cell size. Scale bars in (**A**) and (**B**) = 50 µm. (**C**) Quantification of (**A**) and (**B**) (n = 4–5 independent transplanted animals per genotype; Student's t test). (**D**) Immunohistochemistry for mitochondrial proteins cytochrome oxidase subunit 1 (MT-CO1) and 2 (MT-CO2), transporter of the outer membrane 20 (TOM20) and INS in 3 months old grafts. Quantification of the mean fluorescence intensity of each immunostaining per individual INS+ cell (n > 100 individual INS+ cells per genotype, from 4 to 5 independent transplanted animals per genotype; Student's t test). Scale bars = 20 µm. **p < 0.01, ***p < 0.001.
DOI: https://doi.org/10.7554/eLife.38519.022

cells (*Carrano et al., 2017*). Single-cell RNA-seq has been recently used to study both mouse and human islets in different stages of development and disease (*Baron et al., 2016*; *Segerstolpe et al., 2016*; *Xin et al., 2018*; *Zeng et al., 2017*). These datasets can now be compared with in vitro hPSC-derived beta-cells to verify the identity of the endocrine cells, as we present here (*Figure 2—figure supplement 1B*).

Moreover, single-cell transcriptomic data enables the use of pseudotime analysis to infer differentiation trajectories. This is particularly useful to elucidate hierarchical and temporal relationships between cell types in developing tissues and in vitro differentiation experiments (*Qiu et al., 2017*). Using this approach, we identified a differentiation trajectory with two distinct endocrine progenitor stages, an earlier progenitor population marked by PROX1, HES1 and ID2, and a more differentiated progenitor population marked by CHGA, NKX2.2, FEV, NEUROD1 and MNX1. These two types of progenitors likely represent sequential stages in the differentiation to beta-like cells, which are depicted as independent populations due to the branching trajectory imposed by the analysis algorithm. Interestingly, two putative endocrine progenitor populations have been also observed in a recent study where the stages of hPSC differentiation towards beta-cells were examined by single-cell qRT-PCR (*Petersen et al., 2017*). Further research will be required to determine if these different endocrine progenitor populations have in vivo counterparts in human embryonic pancreas or rather represent an in vitro artifact.

Diabetes caused by misfolded proinsulin has been studied extensively in the Akita mouse model carrying the INS C96Y mutation (*Izumi et al., 2003*; *Oyadomari et al., 2002*; *Wang et al., 1999*) and in the Munich mouse model carrying the INS C95S mutation (*Herbach et al., 2007*). These mice become progressively diabetic 4 to 8 weeks after birth, presenting a reduced beta-cell mass which has commonly been attributed to increased apoptosis (*Oyadomari et al., 2002*). However, significantly increased beta-cell apoptosis was not detected in some of the studies published with these models (*Herbach et al., 2007*; *Izumi et al., 2003*). A closer examination of the Akita mouse postnatal development has revealed reduced proliferation and impaired function of neonatal beta-cells, in the absence of increased apoptosis (Riahi et al, accompanying paper). These findings are well in line with the results of our study, since we could not detect increased apoptosis in the INS mutant cells at any point, despite the elevated ER-stress manifested by the high expression of BIP, MANF and GRP170. A potential caveat of our model is that grafted mice employed in this study were normoglycemic. It remains to be elucidated if additional ER-stress imposed by increased insulin demand in a diabetic environment would eventually lead to increased apoptosis of human INS mutant beta-cells in vivo.

Disruption of mTORC1 signaling is a crucial link between aggravated ER-stress and defective beta-cell expansion in the neonatal period (*Sinagoga et al., 2017*). Inhibition of mTORC1 signaling in *Raptor* KO mouse beta-cells leads to impaired postnatal beta-cell growth, function and mitochondrial function (*Ni et al., 2017*). Restoration of the mTORC1 signaling was sufficient to rescue the beta-cell proliferation defect in Akita mice (Riahi et al., accompanying paper). All these results are in agreement with our findings, where the INS mutant beta-cells show reduced mTORC1 signaling (decreased *LAMTOR5* expression, reduced S6 phosphorylation) and would explain their diminished cell size, altered mitochondria and INS protein content.

ER-stress and mitochondrial function are closely associated in the etiology of diverse diseases. For example, ER-stress signaling pathway component PERK has been shown to regulate mitochondrial morphology (*Lebeau et al., 2018*). In the context of ER-stress induced by misfolded proinsulin, Akita mouse beta-cells present increased mitochondrial dysfunction, with mitochondrial fragmentation and reduced respiration (*Mitchell et al., 2013*). We observed similar mitochondrial defects by scRNA-seq and immunostaining INS mutant beta-cells in vitro and in vivo.

In both rodents and humans, beta-cell proliferation peaks in the neonatal period (*Finegood et al., 1995*; *Gregg et al., 2012*; *Zeng et al., 2017*). A recent scRNA-seq study shows that neonatal proliferative mouse beta-cells are characterized by high mitochondrial membrane potential, expression of immediate early genes *Fos, Egr1, Jun* and *Srf* and increased PI3K-mTOR signaling (*Zeng et al., 2017*). Similar scRNA-seq analysis of human adult beta-cells has shown that beta-cells with high expression of UPR and lower expression of insulin are more prone to proliferate (*Xin et al., 2018*). These cells are more metabolically active, as reflected by their higher expression of glycolytic pathway, tricarboxylic acid cycle and electron transport chain genes.

Interestingly, our scRNA-seq analysis shows downregulation of similar gene sets, including mitochondrial respiratory chain subunits (*MT-CO1, MT-CO2*), immediate early genes (*FOS, EGR1* and *IER*) and *LAMTOR5* in INS C96R beta-like cells (*Figure 2D*, *Figure 2—figure supplement 1C*). Furthermore, basic helix-loop-helix (bHLH) transcription factors *ID1, ID2* and *ID3* were downregulated in the INS C96R cells. These genes, regulated by BMP signaling, are important for the proliferation and differentiation of beta-cells (*Hua and Sarvetnick, 2007*). Taken together, our single-cell

transcriptomic results provided a basis for the reduced proliferation and altered mitochondria of INS mutant beta-like cells, that was later confirmed in vitro and in vivo.

MANF is an ER-stress induced prosurvival factor whose role in reducing UPR activation is critical for mouse beta-cell development and postnatal beta-cell mass expansion and maintenance (*Danilova et al., 2018*; *Lindahl et al., 2014*). *Manf* KO mice presented reduced beta-cell mass and beta-cell proliferation at birth, without increased apoptosis, highlighting the importance of maintaining a physiological level of ER-stress for perinatal beta-cell proliferation. Thus, it is possible that the lack of ER-stress-induced beta-cell apoptosis is in part explained by the high MANF expression triggered by the misfolded proinsulin. Interestingly, we detected increased apoptosis in the INS C96R beta-like cells after inducing additional ER-stress with brefeldin A, but not with other stressors (*Figure 3D*). This difference could be the result of the particular brefeldin A mechanism of action, the inhibition of protein transport from ER to Golgi apparatus, which results in additional ER overloading (*Helms and Rothman, 1992*).

Impaired beta-cell proliferation resulted in skewed endocrine cell proportions with less insulin-positive and more glucagon- and ghrelin-positive cells. Ghrelin-positive cells represent a transient fetal endocrine cell population (*Arnes et al., 2012*), rare in adult human islets (*Segerstolpe et al., 2016*). Mouse adult beta-cells have been shown to lose their identity and misexpress ghrelin upon *Pax6* deletion, together with an expansion of the islet alpha-cell population (*Swisa et al., 2017*). Interestingly, we detected reduced expression of *PAX6 in vitro* and lower PDX1 expression in vivo in the INS mutant cells. Therefore, a small percentage of the cells could have differentiated into alternative cell identities although our results indicate that impaired proliferation and growth is likely to be the main cause for the demise of the INS mutant cells. A potential limitation in the analysis of grafted cells is the extensive ischemia induced cell death upon transplantation (*Faleo et al., 2017*), which could skew cell type proportions and increase graft to graft variability.

Overall, our results indicate that misfolded proinsulin triggers ER-stress concomitantly with *INS* expression, affecting the development of the INS mutant beta-cells by impairing their proliferation without increased apoptosis. Decreased proliferation results in the reduced percentage of INS+ cells observed in vitro and in vivo in the INS mutant cells. Elevated ER-stress leads to reduced mTORC1 signaling and altered mitochondria, which are critical for beta-cell proliferation and function. Importantly, our findings demonstrate that INS mutations leading to neonatal diabetes are already pathogenic during pancreatic development due to failure of neonatal beta-cell expansion. This could theoretically open up new possibilities for the treatment of mutant insulin-associated diabetes through transient stimulation of mTORC1, but this treatment would have to be applied within the neonatal period.

Our study extends the observations from the diabetic Akita model into human diabetes and further emphasizes the role of ER-stress in controlling beta-cell proliferation. These findings may be of relevance for the risk of developing type two diabetes later in life, since the functional beta-cell reserve is established in the perinatal period (*Gregg et al., 2012*; *Meier et al., 2008*).

## Materials and methods

**Key resources table**

| Reagent type (species) or resource | Designation | Source or reference | Identifiers | Additional information |
|---|---|---|---|---|
| Cell line (*Homo sapiens*) Male | HEL71.4 | Biomedicum Stem Cell Center, University of Helsinki | | |
| Cell line (*Homo sapiens*) Male | HEL71.4 - corrected clones A2, F2, F10 and G6 | Biomedicum Stem Cell Center, University of Helsinki | | |
| Cell line (*Homo sapiens*) Female | HEL107.2 | Biomedicum Stem Cell Center, University of Helsinki | | |

*Continued on next page*

*Continued*

| Reagent type (species) or resource | Designation | Source or reference | Identifiers | Additional information |
|---|---|---|---|---|
| Recombinant DNA reagent | CAG-Cas9-T2A-EGFP-ires-puro | DOI: 10.1016/j.celrep.2017.03.055 | Addgene plasmid # 78311 | |
| Antibody | Rabbit anti-OCT4 | | Santa Cruz Biotechnology Cat# sc-9081; RRID:AB_2167703 | ICC; (1:500) |
| Antibody | Mouse anti-TRA1-60 | | Thermo Fisher Scientific Cat# MA1-023; RRID:AB_2536699 | ICC; (1:50) |
| Antibody | Rat anti-SSEA3 | | Millipore Cat# MAB4303; RRID:AB_177628 | ICC; (1:70) |
| Antibody | Rabbit anti-AFP | | Dako Cat# A0008; RRID:AB_2650473 | ICC; (1:500) |
| Antibody | Mouse anti-SMA | | Sigma-Aldrich Cat# A2547; RRID:AB_476701 | ICC; (1:400) |
| Antibody | Mouse anti-TUJ1 | | R and D Systems Cat# MAB1195; RRID:AB_357520 | ICC; (1:500) |
| Antibody | Goat anti-PDX1 | | R and D Systems Cat# AF2419; RRID:AB_355257 | ICC, IHC; (1:200) |
| Antibody | Mouse anti-NKX6.1 | | DSHB Cat# F55A10; RRID: AB_532378 | ICC, IHC; (1:200) |
| Antibody | Rabbit anti-SOX9 | | Millipore Cat# AB5535; RRID:AB_2239761 | ICC; (1:500) |
| Antibody | Sheep anti-NEUROG3 | | R and D Systems Cat# AF3444; RRID:AB_2149527 | ICC; (1:500) |
| Antibody | Guinea pig anti-INS | | Dako Cat# A0564; RRID:AB_10013624 | ICC, IHC; (1:1000) |
| Antibody | Rabbit anti-C-peptide | | Cell Signaling Technology Cat# 4593S; RRID:AB_10691857 | IHC; (1:150) |
| Antibody | Mouse anti-ProINS | | DSHB Cat# GS-9A8; RRID:AB_532383 | IHC; (1:200) |
| Antibody | Mouse anti-GCG | | Sigma-Aldrich Cat# G2654; RRID:AB_259852 | IHC; (1:1000) |
| Antibody | Rabbit anti-CHGA | | Dako Cat# A0564 | IHC; (1:500) |
| Antibody | Rabbit anti-SST | | Dako Cat# A0566; RRID:AB_10013726 | IHC; (1:1000) |
| Antibody | Goat anti-PPY | | Sigma-Aldrich Cat# SAB2500747; RRID:AB_10611538 | IHC; (1:1000) |
| Antibody | Goat anti-GHRL | | Santa Cruz Biotechnology Cat# sc-10368; RRID:AB_2232479 | IHC; (1:300) |
| Antibody | Rabbit anti-KI67 | | Leica Microsystems Cat# NCL-Ki67p; RRID:AB_442102 | IHC; (1:500) |

*Continued on next page*

Continued

| Reagent type (species) or resource | Designation | Source or reference | Identifiers | Additional information |
|---|---|---|---|---|
| Antibody | Mouse anti-PCNA | | Thermo Fisher Scientific Cat# MA5-11358; RRID:AB_10982348 | IHC; (1:200) |
| Antibody | Rabbit anti-BIP | | Cell Signaling Technology Cat# 3177S; RRID:AB_2119845 | IHC; (1:250) |
| Antibody | Rabbit anti-GRP170 | | Abcam Cat# ab124884; RRID:AB_10973544 | IHC; (1:200) |
| Antibody | Goat anti-MANF | | Santa Cruz Biotechnology Cat# sc-34560; RRID:AB_670934 | IHC; (1:300) |
| Antibody | Mouse anti-CDH1 (E-Cadherin) | | BD Biosciences Cat# 610181; RRID:AB_397580 | IHC; (1:500) |
| Antibody | Rabbit anti-pS6 | | Cell Signaling Technology Cat# 4858; RRID:AB_916156 | IHC; (1:400) |
| Antibody | Mouse anti-MT-CO1 | | Abcam Cat# ab14705; RRID:AB_2084810 | IHC; (1:200) |
| Antibody | Rabbit anti-MT-CO2 | | Abcam Cat# ab79393; RRID:AB_1603751 | IHC; (1:100) |
| Antibody | Rabbit anti-TOM20 | | Santa Cruz Biotechnology Cat# sc-11415; RRID:AB_2207533 | IHC; (1:250) |
| Antibody | Rabbit anti-Caspase3 | | Cell Signaling Technology Cat# 9664; RRID:AB_2070042 | IHC; (1:250) |
| Antibody | Mouse Anti-CD184 (CXCR4) Monoclonal Antibody, Phycoerythrin Conjugated, Clone 12G5 | | BD Biosciences Cat# 555974; RRID:AB_396267 | FC; (1:1) |
| Antibody | Mouse IgG2a, kappa Isotype Control, Phycoerythrin Conjugated, Clone G155-178 antibody | | BD Biosciences Cat# 563023 | FC; (1:1) |
| Antibody | Insulin (C27C9) Rabbit Antibody (Alexa Fluor 647 Conjugate) | | Cell Signaling Technology Cat# 9008; RRID:AB_2687822 | FC; (1:80) |
| Antibody | Rabbit IgG Isotype Control (Alexa Fluor 647 Conjugate) antibody | | Cell Signaling Technology Cat# 3452S; RRID:AB_10695811 | FC; (1:40) |
| Antibody | Mouse Anti-NKX6.1 Phycoerythrin Conjugated | | BD Biosciences Cat# 555574 | FC; (1:40) |

*Continued*

| Reagent type (species) or resource | Designation | Source or reference | Identifiers | Additional information |
|---|---|---|---|---|
| Antibody | Mouse IgG1, kappa Isotype Control, Phycoerythrin Conjugated, Clone MOPC-21 antibody | | BD Biosciences Cat# 555749; RRID:AB_396091 | FC; (1:40) |
| Antibody | Mouse Anti-NKX6-1 Alexa Fluor 647 Conjugated | | BD Biosciences Cat# 563338 | FC; (1:40) |
| Antibody | Mouse IgG1 kappa isotype control Alexa 647 Conjugated | | BD Biosciences Cat# 557714; RRID:AB_396823 | FC; (1:40) |
| Antibody | Mouse Anti-PDX1 Phycoerythrin Conjugated | | BD Biosciences Cat# 562161; RRID:AB_10893589 | FC; (1:40) |
| Antibody | Donkey anti-Rabbit IgG (H + L) Highly Cross-Adsorbed Secondary Antibody, Alexa Fluor 350 | | Thermo Fisher Scientific Cat# A10039; RRID:AB_2534015 | IHC; (1:500) |
| Antibody | Donkey anti-Mouse IgG (H + L) Highly Cross-Adsorbed Secondary Antibody, Alexa Fluor 350 | | Thermo Fisher Scientific Cat# A10035; RRID:AB_2534011 | IHC; (1:500) |
| Antibody | Donkey anti-Rabbit IgG (H + L) Highly Cross-Adsorbed Secondary Antibody, Alexa Fluor 488 | | Thermo Fisher Scientific Cat# A-21206; RRID:AB_2535792 | ICC, IHC; (1:500) |
| Antibody | Donkey anti-Mouse IgG (H + L) Highly Cross-Adsorbed Secondary Antibody, Alexa Fluor 488 | | Thermo Fisher Scientific Cat# A-21202; RRID:AB_141607 | ICC, IHC; (1:500) |
| Antibody | Goat anti-Guinea Pig IgG (H + L) Highly Cross-Adsorbed Secondary Antibody, Alexa Fluor 594 | | Thermo Fisher Scientific Cat# A-11076; RRID:AB_2534120 | ICC, IHC; (1:500) |
| Antibody | Donkey anti-Sheep IgG (H + L) Cross-Adsorbed Secondary Antibody, Alexa Fluor 594 | | Thermo Fisher Scientific Cat# A-11016; RRID:AB_2534083 | ICC, IHC; (1:500) |
| Antibody | Donkey anti-Goat IgG (H + L) Cross-Adsorbed Secondary Antibody, Alexa Fluor 594 | | Thermo Fisher Scientific Cat# A-11058; RRID:AB_2534105 | ICC, IHC; (1:500) |
| Antibody | Donkey anti-Goat IgG (H + L) Cross-Adsorbed Secondary Antibody, Alexa Fluor 488 | | Thermo Fisher Scientific Cat# A-11055; RRID:AB_2534102 | ICC, IHC; (1:500) |
| Antibody | Donkey anti-Mouse IgG (H + L) Highly Cross-Adsorbed Secondary Antibody, Alexa Fluor 594 | | Thermo Fisher Scientific Cat# A-21203; RRID:AB_2535789 | ICC, IHC; (1:500) |

*Continued on next page*

*Continued*

| Reagent type (species) or resource | Designation | Source or reference | Identifiers | Additional information |
|---|---|---|---|---|
| Antibody | Donkey anti-Rabbit IgG (H + L) Highly Cross-Adsorbed Secondary Antibody, Alexa Fluor 594 | | Thermo Fisher Scientific Cat# A-21207; RRID:AB_141637 | ICC, IHC; (1:500) |
| Antibody | Goat anti-Rat IgM Heavy Chain Cross-Adsorbed Secondary Antibody, Alexa Fluor 488 | | Thermo Fisher Scientific Cat# A-21212; RRID:AB_2535798 | ICC, IHC; (1:500) |
| Antibody | Donkey anti-Mouse IgG (H + L) Highly Cross-Adsorbed Secondary Antibody, Alexa Fluor 594 | | Thermo Fisher Scientific Cat# A-21203; RRID:AB_2535789 | ICC, IHC; (1:500) |
| Antibody | Donkey anti-Goat IgG (H + L) Cross-Adsorbed Secondary Antibody, Alexa Fluor 594 | | Thermo Fisher Scientific Cat# A-11058; RRID:AB_2534105 | ICC, IHC; (1:500) |
| Sequence-based reagent | CYCLOG qRT-PCR primer pair | | NM_004792 | Fw: TCTTGTCAAT GGCCAACAGAG; Rv: GCCCATCTAAA TGAGGAGTTG; 84 bp |
| Sequence-based reagent | PDX1 qRT-PCR primer pair | | NM_004792 | Fw: TCTTGTCAATGG CCAACAGAG; Rv: GCCCATCTAAAT GAGGAGTTG; 84 bp |
| Sequence-based reagent | NKX6.1 qRT-PCR primer pair | | NM_000209.3 | Fw: AAGTCTACCAA AGCTCACGCG; Rv: CGTAGGCGC CGCCTGC; 52 bp |
| Sequence-based reagent | CHGA qRT-PCR primer pair | | NM_001275.3 | Fw: AACCGCAGAC CAGAGGACCA; Rv: GTCTCAGCCC CGCCGTAGT; 102 bp |
| Sequence-based reagent | INS qRT-PCR primer pair | | NM_020999 | Fw: GACGACGCGA AGCTCACCAA; Rv: TACAAGCTGT GGTCCGCTAT; 98 bp |
| Sequence-based reagent | BIP (HSPA5) qRT-PCR primer pair | | NM_005347.4 | Fw: TGGCTGGAAAG CCACCAAGATGCT; Rv: GGGGGAGGG CCTGCACTTCCAT; 116 bp |
| Sequence-based reagent | sXBP1 qRT-PCR primer pair | | NM_001079539.1 | Fw: CTGCTGAGTC CGCAGCAGGTGCA; Rv: GGTCCAAGTT GTCCAGAATGC; 129 bp |
| Sequence-based reagent | CHOP (DDIT3) qRT-PCR primer pair | | NM_001195053.1 | Fw: GCACCTCCCA GAGCCCTCACTC; Rv: CCCGGGCTGG GGAATGACCA; 120 bp |

*Continued on next page*

*Continued*

| Reagent type (species) or resource | Designation | Source or reference | Identifiers | Additional information |
|---|---|---|---|---|
| Sequence-based reagent | ATF3 qRT-PCR primer pair | | NM_001206488.2 | Fw: AGAAAGAGT CGGAGAAGC; Rv: TGAAGGTTG AGCATGTATATC; 103 bp |
| Sequence-based reagent | ATF4 qRT-PCR primer pair | | NM_001675.2 | Fw: AAGGCGGGC TCCTCCGAATGG; Rv: CAATCTGTCC CGGAGAAGGCATCC; 89 bp |
| Sequence-based reagent | ATF6 qRT-PCR primer pair | | NM_001675.2 | Fw: ACCTGCTGTTA CCAGCTACCACCCA; Rv: GCATCATCACT TCGTAGTCCTGCCC; 120 bp |
| Sequence-based reagent | MANF qRT-PCR primer pair | | NM_006010.4 | Fw: GGCGACTGC GAAGTTTGTAT; Rv: TTGCTTCCC GGCAGAACTTT; 121 bp |
| Sequence-based reagent | GRP170 (HYOU1) qRT-PCR primer pair | | NM_001130991.2 | Fw: GTCCAAGGG CATCAAGGCTC; Rv: TTCTGCGCT GTCCTCTACCA; 103 bp |

## hiPSC derivation

Dermal fibroblasts obtained from a skin biopsy were reprogrammed using retroviral delivery of the OCT4, SOX2, MYC and KLF4 transcription factors, as described elsewhere (*Toivonen et al., 2013*). hiPS cells were cultured on Matrigel (BD Biosciences)-coated plates with E8 medium (Life Technologies, A1517001) and passaged using 5 mM EDTA (Life Technologies, 15575–038) as a dissociation agent. For pluripotency characterization, cells were spontaneously differentiated using embryoid-body assay (*Balboa et al., 2017*). Karyotype analyses based on chromosomal G-banding were performed at Yhtyneet Medix Laboratories, Helsinki, Finland. All hiPSC lines were authenticated using Sanger sequencing for the insulin gene mutations and were negative for mycoplasma contamination test.

## Genome editing

Guide RNAs (gRNAs) targeting the insulin locus were designed using web-based tool http://crispr.mit.edu (*Hsu et al., 2013*), selecting for guide RNAs with high quality scores to avoid possible off-targets. Transcriptional units for gRNA expression were prepared by PCR (*Balboa et al., 2015*) and transfected to HEK293 cells together with WT SpCas9 expressing plasmid CAG-Cas9-T2A-EGFP-ires-puro (Addgene plasmid # 78311). Cutting efficiency was determined using T7 endonuclease I (New England Biolabs) assay (PCR primers: hIns_1229_Fw: GGGTGACCCTCCCTCTAACC, 3'Ins-Rv: TCAGCGGCCGCTCCACAGGGACTCCATCAGA). gRNA Ins8 (CTGGTAGAGGGAGCAGATGC-TGG) was found to cut with high efficiency 9 bp away from the *INS* C96R mutation. A correction strategy was devised, based on the recombination with a 70 bases single stranded DNA oligo (ssODN), complementary to the Ins8 gRNA. This ssODN corrects the C96R mutation and introduces a synonymous coding nucleotide change, disrupting the protospace adjacent motif (PAM) and creating a novel BsrGI restriction site that facilitates the screening of recombinant clones (*Figure 1—figure supplement 2*) (ssODN_Ins8_BsrGI:GCAGAAGCGTGGCATTGTGGAACAATGCTGTACAAGCATCTGCTCCCTCTACCAGCTCGAGAACTACTGC). For correction of the mutation in the patient-derived iPSC, two million HEL71.4 cells were electroporated with 6 μg of CAG-Cas9-T2A-EGFP-ires-puro endotoxin-free plasmid, 500 ng of gRNA-PCR Ins8 product and 6 μg ssODN (Neon Transfection System, 1100 V, 20 ms, two pulses, ThermoFisher). Cells were immediately plated onto Matrigel-coated plates containing E8-medium with 5 μM ROCK inhibitor (Y-27632 2HCl, Selleckchem).

Cells positive for GFP fluorescence were pool-sorted 48 hr later and expanded. Single-cell sorting was performed as previously described (*Saarimäki-Vire et al., 2017*). Plasmids and detailed protocols have been deposited on Addgene (http://www.addgene.org/78311/).

## Differentiation experiments

For differentiation of iPSC to beta-cells a modification of previously published protocols was used (*Pagliuca et al., 2014*; *Rezania et al., 2014*), as described previously (*Saarimäki-Vire et al., 2017*) (*Figure 1—figure supplement 3A*). Cells were dissociated with 5 mM EDTA treatment for 10 min and seeded at 1.5–2 million cells/3.5 cm well on Matrigel-coated plates with E8 medium containing 5 µM ROCK inhibitor (Y-27632 2HCl, Selleckchem). Differentiation was started 24 hr later and proceeded through seven stages differentiation protocol (Stages 1 to 4 in adherent culture and stages 5 to 7 in suspension culture):

- Definitive endoderm induction (3 days): cells were washed 1xPBS (without Ca2+/Mg2+) and 2 mL medium/3.5 cm well of differentiation media was applied: MCDB131 (10372–019, Life Technologies)+2 mM Glutmax (35050038, Life Technologies)+1.5 g/L NaHCO3 (Sigma-Aldrich) + 0.5% BSA fraction V Fatty acid free (Sigma-Aldrich) + 10 mM final glucose (Sigma-Aldrich) + 100 ng/ml ActA + 3 µM CHIR. CHIR concentration was reduced to 0.3 µM and 0 µM on day 1 and day two respectively.
- Stage 2 posterior foregut induction (for 3 days, medium changed every day): MCDB131 + 2 mM Glutmax + 1.5 g/L NaHCO3 + 0.5% BSA fV + 10 mM final glucose + 0.25 mM Ascorbic acid (A4544, Sigma-Aldrich)+50 ng/mL FGF7 (Z03047, Genscript)
- Stage 3 pancreatic endoderm induction (for 2 days, medium changed every day): MCDB131 + 2 mM Glutmax + 2.5 g/L NaHCO3 + 2% BSA fV + 10 mM final glucose + 0.25 mM Ascorbic acid + 50 ng/mL FGF7 + 0.25 µM SANT1 (S4572, Sigma-Aldrich)+1 µM RA + 100 nM LDN + 1:200 ITS-X (51500–056, Life Technologies)+200 nM TPB (sc-204424, Santa Cruz).
- Stage 4 pancreatic progenitor induction (3 days, medium changed every day): MCDB131 + 2 mM Glutmax + 2.5 g/L NaHCO3 + 2% BSA fV + 10 mM final glucose + 0.25 mM Ascorbic acid + 2 ng/mL FGF7 + 0.25 µM SANT1 + 0.1 µM RA + 200 nM LDN + 1:200 ITS-X + 100 ng/mL EGF (AF-100–15, Peprotech)+10 mM Nicotinamide (N0636, Sigma-Aldrich).
- Stage 5 endocrine induction: Cells were washed 2x with EDTA, treated with TrypLE for 10 min at 37°C, 1 mL/3.5 cm well. Cells were dissociated by pipetting, and spin down for 3 min at 200 rcf. Cells were resuspended in S5 medium with ROCK inhibitor at 10 µM, at a density of one million cells/mL, in a volume of 5 mL, in ultra-low attachment 6-well plates (3471, Corning) and placed in a rotating platform (InforsHT, Celltron) at 95 rpm in the incubator. After 24 hr, cells formed round aggregates and medium was changed to S5 without ROCK inhibitor: S5 medium (for 4 days, changed every day): MCDB131 + 2 mM Glutmax + 1.5 g/L NaHCO3 + 2% BSA fV + 20 mM final Glucose + 1:200 ITS-X + 10 ug/mL Heparin (H3149, Sigma-Aldrich)+0.25 µM SANT1 + 0.05 µM RA + 100 nM LDN + 10 µM ALK5inhII (S7233, Selleckchem)+1 µM GC1 (4554, Tocris)+10 µM Zinc Sulfate (Z0251, Sigma-Aldrich)+20 ng/mL Betacellulin (100–50, Peprotech)+100 nM GSiXX (565789, Millipore).
- Stage 6 media (14 days, change every second day): MCDB131 + 2 mM Glutmax + 1.5 g/L NaHCO3 + 2% BSA fV + 20 mM final glucose + 1:200 ITS-X + 10 µg/mL Heparin + 100 nM LDN + 10 µM ALK5inhII + 1 µM GC1 + 100 nM GSiXX + 10 µM Zinc Sulfate.
- Stage 7 media (7 days, change every second day): MCDB131 + 2 mM Glutmax + 1.5 g/L NaHCO3 + 2% BSA fV + 20 mM final glucose + 1:200 ITS-X + 10 µg/mL Heparin + 10 µM ALK5inhII + 1 µM GC1 + 10 µM Trolox + 1 mM N-Acetylcysteine (A9165, Sigma-Aldrich)+75 µM Resveratrol (R5010, Sigma-Aldrich)+20 µM JNK inhibitor (SP600125, #1496, Tocris)+2 µM R428 (S2841, Selleckchem).

To improve reproducibility and standardize the differentiation, small molecule compounds were prepared in batches of stage-specific supplements, enabling rapid differentiation media preparation and consistency between experiments:

## S3 supplement preparation, 2500X stock

SANT1 (2.5 mM) 100 µL 200 µL
 RA (10 mM) 100 µL 200 µL
 LDN (1 mM) 100 µL 200 µL

TPB (2.5 mM) 80 µL 160 µL
DMSO 20 µL 40 µL
TOTAL 400 µL 800 µL

S4 supplement preparation, 2500X stock
LDN (1 mM) 200 µL 400 µL
SANT1 (2.5 mM) 100 µL 200 µL
TPB (2.5 mM) 40 µL 80 µL
RA (10 mM) 10 µL 20 µL
DMSO 50 µL 100 µL
TOTAL 400 µL 800 µL

S5 supplement preparation, 1500X stock
RA (10 mM) 5 µL 10 µL
SANT1 (2.5 mM) 100 µL 200 µL
LDN (1 mM) 100 µL 200 µL
GC1 (10 mM) 100 µL 200 µL
GSiXX (1 mM) 100 µL 200 µL
ALK5inhII (50 mM) 200 µL 400 µL
DMSO 61.6 µL 123.2 µL
TOTAL 666.6 µL 1333.2 µL

S6 supplement preparation, 2000X stock
ALK5inhII (50 mM) 200 µL 400 µL
LDN (1 mM) 100 µL 200 µL
GC1 (10 mM 100 µL 200 µL
GSiXX (1 mM) 100 µL 200 µL
TOTAL 500 µL 1000 µL

## Flow cytometry analysis

Cytometry for definitive endoderm marker CXCR4+ was performed as previously described (*Saari-mäki-Vire et al., 2017*). For intracellular antigen pancreatic marker cytometry of Stage 4 or Stage 7, cells were dissociated with TrypLE for 5–10 min at 37°C and resuspended in 5% FBS-containing PBS. Cells were fixed and permeabilized using Cytofix/Cytoperm (554714, BD Biosciences) as recommended by manufacturer. Primary or conjugated antibodies were incubated with the cells overnight at 4°C in Perm/Wash buffer (554714, BD Biosciences) containing 4% FBS. Cells were washed 2x with Perm/Wash buffer and analysed using FACSCalibur cytometer (BD Biosciences) and FlowJo software (Tree Star Inc.).

## Quantitative RT-PCR

Total RNA was isolated using NucleoSpin Plus RNA kit (Macherey-Nagel). SimpliNano (General Electric) spectrophotometer was used to measure RNA quality and concentration. A total of 1.5 µg RNA was denatured at 65° C for 1 min and reverse transcribed (RT) with 0.5 µL Moloney murine leukemia virus (MMLV) reverse transcriptase (M1701, Promega), 0.2 µL Random Primers (C1181, Promega), 1 µL Oligo(dT)18 Primer (SO131, ThermoFisher) and 0.5 µL Ribolock RNAse inhibitor (EO0382, ThermoFisher) for 90 min at 37° C. qRT-PCR reactions were prepared with 50 ng of retrotranscribed RNA were amplified with 5 µL of forward and reverse primer mix at 2 µM each using 5x HOT FIREPol Eva-Green qPCR Mix Plus (no ROX) in a final volume of 20 µL. QIAgility (Quiagen) liquid handling system was used for pipetting the reactions into 100 well disc that were subsequently sealed and run in Rotor-Gene Q (Qiagen) with a thermal cycle of 95° C for 15 min, followed by 40 cycles of 95° C, 25 s; 57° C, 25 s; 72° C, 25 s, followed by a melting step.

Relative quantification of gene expression was analysed using $\Delta\Delta$Ct method, with cyclophilin G (PPIG) as endogenous housekeeping control gene. RT-reaction without template was used as negative control and exogenous positive control was used as a calibrator.

Expression levels were normalized in each sample by the percentage of INS+ cells determined by cytometry and presented as relative to INS corrected cells. See Key Resource Table for primer list.

## Single-cell RNA sequencing

### Single-cell transcriptome barcoding in droplets and library preparation for Illumina sequencing

For scRNAseq, InDrop (*Klein et al., 2015*) was implemented following the protocol as previously described (*Zilionis et al., 2017*). Briefly, stage 7 islet-like aggregates were dissociated into single cells by incubation with a 1:1 mixture of TrypLE Express and Trypsin-EDTA for 10 min at 37°C. Dissociated cells were passed through a 30 µm strainer to remove cell clumps. Single cells were co-encapsulated into 3–4 nL droplets together with barcoded hydrogel beads and a mixture of reverse-transcription (RT) and lysis reagents. Within every single droplet, a cell was lysed and cDNA tagged with a barcode during reverse transcription. The droplet emulsion was broken and the bulk material was taken through the following steps: i) second strand synthesis; ii) linear amplification by in vitro transcription (IVT); amplified RNA fragmentation; iv) reverse transcription; v) PCR. In total, four samples from independent experiments were processed. They were produced in two parallel differentiation runs from HEL71.4 mutant and HEL71.4-A2 corrected iPSC lines, that were encapsulated at two different timepoints. The resulting DNA libraries were multiplexed and sequenced together on Next-Seq Illumina platform in paired-end mode using a high-yield 75 cycle kit. Read quality was assessed by running FASTQC (version 0.10.1).

### Sequencing read processing

To process the sequencing reads and to generate count matrices, a previously published set of Python scripts were used (*Klein et al., 2015*; *Zilionis et al., 2017*) (Supplementary materials). Briefly, raw transcript reads were trimmed with Trimmomatic 0.36, and barcode reads were filtered for expected structure (known cellular barcode, W1 sequence, poly-T tail). Modifications to the original scripts include a hamming distance of 5 for W1 sequence matching and minimum poly-T length of 3 for the barcode reads. Filtered reads were sorted based on barcodes, and barcodes having at least 20 000 reads were kept. The filtered reads were aligned with bowtie 1.1.1 (*Langmead et al., 2009*) to the Ensembl GRCh38 cDNA reference from which haplotypic transcripts were removed. A poly-A sequence of 125 bases was added to each transcript before building the bowtie index. Details on read alignment and UMI quantification arguments can be found in the supplementary materials. Percentages of unaligned reads per barcode were recorded from the script output and used in count matrix quality control. See also *Supplementary file 1* – Table 9.

### Count matrix quality control

We examined the percentages of unmapped reads, total numbers of counts, detected genes per cell, and percentages of apoptotic and mitochondrial counts of cells, and removed the outliers based on these metrics. Cells with less than 50% unmapped reads were kept. Cells with total numbers of counts and detected genes three median-absolute-deviations above or below the median ($\log_{10}$ scale) in a given sample were removed. Cells with over 2% apoptosis gene counts (3 out of 1295 mutant and 2 out of 992 corrected cells) were filtered out. As apoptosis genes, the HALLMARK_APOPTOSIS gene set from Broad Institute's Molecular Signatures Database (MSigDB) was used. The distributions of mitochondrial counts per cell were different between genotypes: the proportions were higher for corrected samples and the upper tail of their distribution extended further (*Figure 2—figure supplement 3*). The thresholds for filtering cells were adjusted accordingly: 20% for mutant cells, and 40% for corrected cells (corresponding to values 3.5 standard deviations above the mean, 22 out of 1295 mutant cells and 17 out of 992 corrected cells were removed based on their mitochondrial gene expression)(*Figure 2—source data 2*). The R package scater 1.4.0 was used for quality control of count data. See also *Supplementary file 1* – Table 9.

### Clustering

Clustering was performed with Seurat 2.0.1 (Satija, R., Butler, A. and Hoffman, P., 2017. Seurat: Tools for Single Cell Genomics. R package version 2.0.1. https://CRAN.R-project.org/package= Seurat). The samples were divided into genotype groups and normalized regressing out the sample

identity and total number of counts (nUMI). The genotype groups were combined using Seurat's integration strategy in which datasets are aligned based on their shared gene correlation structure (*Butler et al., 2018*). Canonical correlation analysis (CCA) was performed on the data sets to identify their common sources of variation. For CCA, the union of the 1000 most highly variable genes in the two data sets was used. The first 8 CCA dimensions were used to align the mutant and corrected samples with Seurat's AlignSubspaces function. The cells were clustered in aligned CCA space (eight dimensions) with Seurat's FindClusters function using a resolution of 0.8 and the SLM algorithm. Sample identity and nUMI were passed as latent variables to the clustering function. Clusters that had less than 20 differentially expressed (DE) genes with a fold change of 1.5 (FDR = 5%) between them were merged. DE genes for all pairs of clusters were identified with Seurat's function FindMarkers with the following arguments: min.pct = 0, test.use = 'negbinom', thresh.use = log(1.5), only.pos = F, latent.vars = c('sample','nUMI').

Cluster robustness was evaluated using a resampling procedure following *Joost et al. (2016)* (*Joost et al., 2016*). Briefly, a thousand random subsamples (75% of the cells before cluster merging and refinement) were selected without replacement and clustered with arguments equivalent to the original run. For each original cluster, the maximum proportion of cells co-clustered in each subsample was recorded. The same procedure was repeated for shuffled cluster-cell pairs, to determine the null distribution of co-clustering proportions.

The cells were mapped to the adult human pancreas data of *Baron et al. (2016)* using the R package scmap 1.0.0 (*Kiselev et al., 2018*). The *Baron et al. (2016)* data were downloaded from https://hemberg-lab.github.io/scRNA.seq.datasets/ and used to create an scmap reference. The reference was created using 250 genes selected with scmap's getFeatures function from the genes common to both data sets. For projecting cells onto the reference, a similarity threshold of 0.6 was used. Cells in the beta-like and progenitor clusters mapping to alpha cells were moved to the alpha-like cluster. Cells mapping to delta and ductal cells (25 and 1 cells, respectively) were removed from the data set.

Cluster markers were identified using Seurat's function FindAllMarkers with the following arguments: thresh.use = log(1.5), test.use = 'negbinom', min.pct = 0, latent.vars = ('sample','nUMI'), only.pos = T, return.thresh = 1. The p-values of the tests were adjusted for multiple testing with the Benjamini-Hochberg method.

Dimensionality reduction with t-SNE was performed using Seurat's RunTSNE function with the following arguments: reduction.use = 'cca.aligned', dims.use = 1:8, do.fast = T, perplexity = 70.

## Differential expression between mutant and corrected cells

For cells in the refined beta-like and progenitor clusters, differential expression between mutant and corrected cells was tested with MAST 1.4.0 (*Finak et al., 2015*). The expression data used were log-normalized values (Seurat default) with base changed to 2. Only genes expressed in at least 3 cells in both genotype groups were considered. A hurdle model was fitted with terms for genotype and encapsulation day, and the genotype effect was tested with a likelihood ratio test. P-values were adjusted for multiple testing using the Benjamini-Hochberg method. Genes with fold changes of at least 1.3 and adjusted p-values below 0.05 were considered differentially expressed.

## Functional enrichment analysis

The statistical overrepresentation test for GO terms was performed with PANTHER at www.pantherdb.org. The binomial test was run with default settings using lists of gene names, the biological process (BP) category, and all human genes in the database as reference. Bonferroni correction for multiple testing was used. GSEA was performed with CAMERA of the R package limma 3.32.6. The curated C2 gene sets were downloaded from MSigDB. The same gene expression values and linear model formula were used for GSEA and differential expression tests with MAST. Only genes expressed in at least 3 cells in both genotype groups were considered. An FDR of 5% was used as the threshold for statistical significance.

## Pseudotime analysis

Pseudotime analysis was performed with the R package Monocle 2.4.0 (*Qiu et al., 2017*). The mutant and corrected samples were processed and ordered together. Highly variable genes used

for ordering (with empirical dispersion at least two times greater than the dispersion fit) were selected from the corrected samples only, to avoid biasing the ordering by the mutant cells. Dimensionality of the data was reduced with the DDRTree algorithm.

Sample identity was used in the residual model formula to reduce sample-specific effects on ordering. This was done to enable side-by-side comparison of mutant and corrected cells on the same trajectory. Branch dependent genes, for examining the differences between progenitor branches, were identified with Monocle 2's BEAM function controlling for sample identity. Pseudo-time was reversed by applying

$$max(P) - p,$$

where $P$ is the set of pseudotime values, to each pseudotime value $p$.

Pseudotime-dependent genes were identified with Monocle 2's differentialGeneTest function. The test was performed separately for mutant and corrected samples controlling for sample identity. The union of pseudotime-dependent genes (q-value < 0.05%) of the mutant and corrected samples were further used for testing e differentially expressed genes between genotypes along pseudotime. This test was performed with Monocle 2's function differentialGeneTest using the combined data, controlling for encapsulation date and pseudotime. A q-value of 0.05 was set as the threshold of statistical significance. Scripts used for these analyses can be found in the Supplementary materials.

### Induction of apoptosis by ER-stress inducers
One hundred manually picked Stage 7 islet-like aggregates were incubated in full Stage 7 media with the corresponding concentration ER-stress inducers. Brefeldin A (B5936, Sigma-Aldrich) was used at 1 µg/mL in DMSO for 24 hr. Thapsigargin (T9033, Sigma-Aldrich) and tunicamycin (T7765, Sigma-Aldrich) were used at 1 µM and 5 µg/mL respectively for 48 hr. DMSO was used as a vehicle control at 5 µL/mL. Aggregates were collected and PFA-fixed for immunohistochemistry after treatment.

### Static glucose stimulated insulin secretion
Stage 7 islet-like aggregates were sampled in groups of 100 to 1.5 mL tubes. They were washed twice with Krebs buffer containing no glucose and then transferred to 12-well plate placed in a rotating platform for incubation in 3.3 mM glucose-containing Krebs buffer for 1 hr (low glucose). This was performed twice. Then aggregates were incubated sequentially in 3.3 mM glucose, 20 mM glucose, 20 mM glucose + 100 µM tolbutamide and 3.3 mM + 30 mM KCl, for a period of 30 min, with two washes with 1 mL Krebs buffer containing no glucose between treatments. 500 µL of supernatant from each treatment incubation were collected, centrifuged to remove possible cells in suspension and stored at −80°C for ELISA-based determination of human insulin concentration. After the last treatment incubation, samples were retrieved and lysed in acid ethanol for determination of total insulin content and DNA content. Stimulated insulin secretion results are presented as fractional release of total human insulin content after cell mass normalization using total DNA content.

### Transplantation of differentiated cells
NOD-SCID-gamma (NSG) (Jackson Laboratories; 005557) mice were housed at Biomedicum Helsinki animal facility, on a 12 hr light/dark cycle and food ad libitum. Transplantations were performed on 3- to 12-month- old mice as described previously (*Saarimäki-Vire et al., 2017*). Briefly, aggregates equivalent to approximately 5 million cells were loaded on a PE-50 tubing and transplanted under the kidney capsule. Mice were anesthetized with isoflurane. Carprofen (Rimadyl, 5 mg/kg, subcutaneously, Prizer, Helsinki, Finland) and Buprenorphine (Temgesic, 0,05–0,1 mg/kg, subcutaneously, RB pharmaceuticals Lmt, Berkshire, UK) were used as analgesics during the operation and in the following day. Mouse blood samples were collected monthly from the saphenous vein using heparinized capillary tubes. Blood plasma was separated by centrifugation (5000 rcf, 5 min, RT).

### Intraperitoneal glucose tolerance test (IPGTT)
IPGTT was performed after 6–8 hr fast. 2 g glucose/kg of body weight was injected intraperitoneally in the form of a 30% glucose solution in water. Blood glucose levels were measured with glucometer (OneTouch Ultra, Lifescan, Milpitas; USA) at 0, 20, 40 and 60 min after glucose injection. Blood samples for measuring human c-peptide levels were taken before, and 40 min after glucose injection.

## ELISA

Human c-peptide and proinsulin levels were measured from plasma samples and cell supernatants with Ultrasensitive C-PEPTIDE ELISA (Mercodia, Sweden) and PRO-INSULIN (Mercodia, Sweden) according to manufacturer's instructions. Human MANF levels were measured using in-lab ELISA (Galli et al. 2016).

## Immunocytochemistry and histology

Cells on adherent cultures were fixed in 4% PFA for 15–20 min, permeabilized with 0.5% triton-X100 in 1x PBS, blocked with UltraV block (ThermoFisher) for 10 min and incubated with primary antibodies diluted in 0.1% Tween in 1 x PBS at 4°C overnight. Cells were washed with 1 x PBS, incubated with secondary antibodies diluted in 0.1% Tween in PBS. Same procedure was used for whole-mount staining of Stage 7 cell aggregates. For paraffin embedding, Stage 7 cell aggregates were fixed with 4% PFA at RT overnight and briefly stained with Eosin. After this, they were embedded in low-melting Agarose (Sigma-Aldrich) and transferred to paraffin blocks. Grafts were retrieved after IPGTT, dissected and fixed with 4% PFA in RT overnight and placed into cassettes and processed for tissue transfer and paraffin embedding. Paraffin blocks were cut into 5 μm sections. For immunohistochemistry, slides were deparaffinised and antigen retrieval was performed by boiling slides either in 1 mM EDTA or 0.1 M citrate buffer. Blocking and incubation with primary and secondary antibodies were done as described for fixed cells above. For TUNEL analysis, paraffin sections were processed with In Situ Cell death Detection Fluorescein kit (Roche, #11684795910) according to manufacturer's instructions. See Key Resource Table for list of antibodies (ICC: immunocytochemistry on fixed cells; IHC: immunohistochemistry on paraffin sections; FC: flow cytometry)

## Image acquisition and analysis

Immunofluorescence stainings of adherent cells were imaged with EVOS inverted microscope (Life-Technologies). Paraffin sections and whole-mount stainings were imaged with Zeiss Axio Observer Z1 with Apotome two and processed with ZEN2 software blue edition. To ensure reliable quantification of the immunostainings, all paraffin sections were stained simultaneously and imaged on the same session with the same microscope parameters. Image quantifications were performed blindly using Fiji software (*Schindelin et al., 2012*). Quantification of individual cell immunostaining intensity was performed manually using Fiji ROI Manager and Multi Measure tools. Fiji pixel intensity and Cell counter tools were used to score percentages of cells positive, negative, low or high across the different immunostainings.

## Statistical analyses

Statistical analyses were performed with GraphPad Prism (version 7.0 c, GraphPad Software). Data were tested for normal distribution using Shapiro-Wilk normality test. Normally distributed data were analyzed to compare the means of two samples using unpaired two-tailed Student's t test, with Welch's correction in the case of samples with unequal variance as determined by F test. One-way ANOVA with multiple comparison Tukey test was used to compare the means of more than two samples. When the data groups were not normally distributed or the sample size was too small, the non-parametric Mann-Whitney U test and Kruskal-Wallis test with the Dunn multiple comparisons test were used to compare the sum of ranks. Details on the statistical analyses performed are described in the figure legends. Data are presented as individual value points and/or the mean as summary statistic with error bars representing the Standard Error of the Mean (SEM). P-values under 0.05 were considered statistically significant (*$p < 0.05$, **$p < 0.01$, ***$p < 0.001$).

## Study approval

The Coordinating Ethics Committee of the Helsinki and Uusimaa Hospital District (no. 423/13/03/00/08) approved the patient informed consent for the derivation of the hiPSC lines used in this study: HEL71.4 and HEL107.2.

Animal care and experiments were approved by National Animal Experiment Board in Finland (ESAVI/9978/04.10.07/2014).

## Acknowledgements

We thank the people carrying the insulin mutations for providing skin biopsies and consent to derive iPSC. Jaan Palgi, Eila Korhonen, Anni Laitinen, Väinö Lithovius, Hazem Ibrahim, Anna Näätänen, Biomedicum Helsinki Animal Facility and FACS Core Facility are thanked for their technical assistance. We thank Riikka Äänismaa for sharing antibodies against mitochondrial proteins. We are grateful to Päivi Miettinen, Anu Suomalainen-Wartiovaara, Henna Tyynismaa, Thomas McWilliams, Tom Barsby, Yuval Novik, Jere Weltner, Emilia Kuuluvainen, Elena Senís, Mariana Igolillo-Steve, Miriam Cnop and Decio Eizirik for discussion and comments on the manuscript. We thank Simon Joost for helping to implement the clustering robustness procedure, and Laura Mäenpää for reviewing the R scripts. DB is a member of the Doctoral School of Health Sciences at University of Helsinki. This project was funded by the Academy of Finland, Sigrid Jusélius Foundation, Novo Nordisk Foundation, the EU 7FP Integrated project BETACURE, the Diabetes Research Foundation, Diabetes Wellness Foundation grants to DB and JS (598–145F), and the Biomedicum Helsinki Foundation and Maud Kuistila Memorial Foundation grants to DB. The project has also received funding from the Innovative Medicines Initiative 2 Joint Undertaking under grant agreement No 115797 (INNODIA), which receives support from the European Union's Horizon 2020 research and innovation programme and 'EFPIA', 'JDRF' and 'The Leona M. and Harry B. Helmsley Charitable Trust'.

## Additional information

### Funding

| Funder | Grant reference number | Author |
|---|---|---|
| Diabetes Wellness Foundation | 598–145F | Diego Balboa<br>Jonna Saarimäki-Vire |
| Biomedicum Helsinki-säätiö | | Diego Balboa |
| Maud Kuistilan Muistosäätiö | | Diego Balboa |
| Suomen Akatemia | | Timo Otonkoski |
| Sigrid Juséliuksen Säätiö | | Timo Otonkoski |
| Novo Nordisk Fonden | | Timo Otonkoski |
| European Commission | EU 7FP Integrated project BETACURE | Timo Otonkoski |
| Diabetesliitto | 598–145F | Timo Otonkoski |
| Innovative Medicines Initiative | 115797 | Timo Otonkoski |

The funders had no role in study design, data collection and interpretation, or the decision to submit the work for publication.

### Author contributions

Diego Balboa, Conceptualization, Formal analysis, Supervision, Investigation, Visualization, Methodology, Writing—original draft, Project administration, Writing—review and editing; Jonna Saarimäki-Vire, Conceptualization, Formal analysis, Investigation, Visualization, Methodology, Writing—original draft, Writing—review and editing; Daniel Borshagovski, Data curation, Software, Formal analysis, Investigation, Visualization, Methodology, Writing—original draft, Writing—review and editing; Mantas Survila, Formal analysis, Investigation, Methodology, Writing—original draft, Writing—review and editing; Päivi Lindholm, Investigation, Writing—review and editing; Emilia Galli, Investigation; Solja Eurola, Jarkko Ustinov, Heli Grym, Investigation, Methodology; Hanna Huopio, Resources; Juha Partanen, Resources, Writing—review and editing; Kirmo Wartiovaara, Conceptualization, Funding acquisition, Investigation, Methodology, Writing—review and editing; Timo Otonkoski, Conceptualization, Resources, Supervision, Funding acquisition, Writing—original draft, Project administration, Writing—review and editing

Author ORCIDs
Diego Balboa (iD) http://orcid.org/0000-0002-4784-5452
Päivi Lindholm (iD) http://orcid.org/0000-0003-3022-5035
Emilia Galli (iD) http://orcid.org/0000-0002-7419-611X
Timo Otonkoski (iD) http://orcid.org/0000-0001-9190-2496

### Ethics

Human subjects: The Coordinating Ethics Committee of the Helsinki and Uusimaa Hospital District (no. 423/13/03/00/08) approved the patient informed consent for the derivation of the hiPSC lines used in this study: HEL71.4 and HEL107.2.
Animal experimentation: Animal care and experiments were conducted as approved by the National Animal Experiment Board in Finland (ESAVI/9978/04.10.07/2014).

### Decision letter and Author response

Decision letter https://doi.org/10.7554/eLife.38519.030
Author response https://doi.org/10.7554/eLife.38519.031

## Additional files

### Supplementary files

• Supplementary file 1. Table 1: Predicted Ins8 gRNA off-target sites assessed for mutations by Sanger sequencing. Table 2: Single-cell RNA-seq analysis cluster marker genes. Table 3: Differentially expressed genes in beta-like cluster between INS C96R vs INS corrected cells. Table 4: Differentially expressed genes in progenitor cluster between INS C96R vs INS corrected cells. Table 5: Gene Set Enrichment Analysis of INS C96R vs INS corrected cells. Table 6: Gene Ontology Analysis of INS C96R vs INS corrected cells. Table 7: Differentially expressed genes between pseudotime analysis progenitor branches. Table 8: Differentially expressed genes along pseudotime between INS C96R vs INS corrected cells. Table 9: Single-cell RNA-seq reads and quality control statistics.
DOI: https://doi.org/10.7554/eLife.38519.023

• Source code 1. Python and R scripts used in the analysis of the single-cell data in this manuscript.
DOI: https://doi.org/10.7554/eLife.38519.024

• Transparent reporting form
DOI: https://doi.org/10.7554/eLife.38519.025

### Data availability

Single cell RNA sequencing raw data was deposited in GEO under GSE115257 Source data for single cell RNA sequencing as well as code scripts for analysis have been provided.

The following dataset was generated:

| Author(s) | Year | Dataset title | Dataset URL | Database and Identifier |
|---|---|---|---|---|
| Balboa D, Borshagovski D, Survila M | 2018 | The raw single-cell RNA sequencing data used in the study | https://www.ncbi.nlm.nih.gov/geo/query/acc.cgi?acc=GSE115257 | NCBI Gene Expression Omnibus, GSE115257 |

The following previously published dataset was used:

| Author(s) | Year | Dataset title | Dataset URL | Database and Identifier |
|---|---|---|---|---|
| Veres A, Baron M | 2016 | A single-cell transcriptomic map of the human and mouse pancreas reveals inter- and intra-cell population structure | https://www.ncbi.nlm.nih.gov/geo/query/acc.cgi?acc=GSE84133 | NCBI Gene Expression Omnibus, GSE84133 |

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
