## [Decision Letter]

Thank you for submitting your article "Insulin mutations impair β-cell development in a patient-derived iPSC model of neonatal diabetes" for consideration by *eLife*. Your article has been reviewed by three peer reviewers, and the evaluation has been overseen by a guest Reviewing Editor and Harry Dietz as the Senior Editor. The following individual involved in review of your submission has agreed to reveal his identity: Holger Russ. Two other reviewers remain anonymous.

The reviewers have discussed the reviews with one another and the Reviewing Editor has drafted this decision to help you prepare a revised submission.

The authors describe the phenotype of iPS-derived β-cell cells with a mutations in the insulin gene which cause permanent neonatal diabetes. These mutations impairs insulin folding and were previously thought to exert most of their deleterious effects via an increase in programmed cell death. However, in this work the authors demonstrate that the break on β-cell proliferation caused by ER stress is a major contributor to the phenotype. This is an important observation and follows on an increasing understanding that β-cells are unusual cells with regards to their baseline ER stress state and how they compensate for ER stress levels. This is a stellar paper and a major contribution to the field. The conclusions are straightforward and well supported by data. The isogenic corrected controls allow them to draw strong conclusions about the role of the specific mutations that are independent of the genetic background. The single-cell sequencing provides an unbiased profile of the cells and the transplantation experiment provides in vivo context.

Despite the authors employing a state-of-the art approach to their study there are limitations with the current methodology which have not currently been adequately addressed in the Discussion. The reviewers and handling editor encourage the authors to be more explicit in the current limitations with IPS cell derived models. This should not be seen as a criticism of the study, more a reflection of the reality of the relative infancy of the approach. The reviewers also recognise that the accompanying manuscript reporting the Akita Mouse model complements the results and in combination the two studies present a compelling mechanism.

Essential revisions:

There are concerns that some of the limitations of the in vivo maturation of the islets in mice have not been acknowledged and that the potential confounding factors have not been adequately discussed. The authors are encouraged to ensure that the following critical points are conveyed to readers.

1) There are concerns that a large portion of transplanted cells could be lost within the first few days in vivo. The authors should comment on whether the mutant and corrected β like cells (and other hormone expressing cells) present at similar percentages immediately after transplant. It seems that at 1 month post-transplant the percentage of INS +ve cells.

2) The reviewers are not in agreement with the conclusion that the increase in circulating human c-peptide in corrected grafts is due to maturation of β-like cells, but would argue that this could be due to the differentiation of β-like cells from co-transplanted progenitors. There is a significant increase after 2-5 months, a time frame that has previously shown to correlate with de novo generation of hormone expressing cells from hPSC derived progenitors upon transplant.

3) The authors evaluate proliferation in β-like cells after 1 and 6 months in vivo, but do not find significantly reduced proliferation in mutant cells. However, graft numbers are rather low to confidently asses this key point of the study and I would encourage the authors to examine more mice. There appears to be a trend towards reduced proliferation but this is at present hard to interpret since the percentage of Ki67 +ve cells is highly variable between individual grafts. This critical point should be examined in more detail.

4) The authors should acknowledge that the β-like cells are not functional in vitro and in vivo, they do not respond with increases in insulin secretion to a glucose challenge (also depolarization shows only modest secretion). It therefore impossible to draw strong conclusions as to the effects of the mutations on mature β cell physiology. The authors are encouraged to discuss this limitation.

---

## [Author Response]

Essential revisions:There are concerns that some of the limitations of the in vivo maturation of the islets in mice have not been acknowledged and that the potential confounding factors have not been adequately discussed. The authors are encouraged to ensure that the following critical points are conveyed to readers.

We have now added discussion of the limitations of our approach. In the first paragraph of the discussion we stress the fact that the stem-cell derived β cells are very immature and that the differentiation efficiency is highly variable between lines.

1) There are concerns that a large portion of transplanted cells could be lost within the first few days in vivo. The authors should comment on whether the mutant and corrected β like cells (and other hormone expressing cells) present at similar percentages immediately after transplant. It seems that at 1 month post-transplant the percentage of INS +ve cells are higher in mutant cell lines.

Our results show that the percentage of INS+ cells is reduced in INS mutant S7 islet-like aggregates already in vitro. We have generated and examined additional 1 month grafts and pooled them with the previous data. There are no significant differences in the% INS+/CHGA+ at 1 month (Figure 2—figure supplement 4A), but the variability is quite high. These results likely reflect the difficult-to-control variability of differentiation efficiency and graft outcome. Furthermore, stem cell derived islet-like cells may suffer from ischemia-induced cell death after transplantation. This has been reported by Faleo, Russ et al., 2017, where they describe up to 40% of graft mass loss 7 days after transplantation under the kidney capsule. This could as well affect the graft cell composition and the graft to graft variability. We have highlighted these limitations in the Discussion, tenth paragraph.

2) The reviewers are not in agreement with the conclusion that the increase in circulating human c-peptide in corrected grafts is due to maturation of β-like cells, but would argue that this could be due to the differentiation of β-like cells from co-transplanted progenitors. There is a significant increase after 2-5 months, a time frame that has previously shown to correlate with de novo generation of hormone expressing cells from hPSC derived progenitors upon transplant.

We agree with the reviewers that the increase in circulating human c-peptide could also be explained by the differentiation of co-transplanted progenitors into additional β-like cells. We have modified the text throughout the manuscript accordingly to reflect this possibility (particularly in the subsection “INS mutant beta-like cells presented reduced insulin secretion after in vivo transplantation”).

3) The authors evaluate proliferation in β-like cells after 1 and 6 months in vivo, but do not find significantly reduced proliferation in mutant cells. However, graft numbers are rather low to confidently asses this key point of the study and I would encourage the authors to examine more mice. There appears to be a trend towards reduced proliferation but this is at present hard to interpret since the percentage of Ki67 +ve cells is highly variable between individual grafts. This critical point should be examined in more detail.

We have been able to produce a few additional 1 month transplants within the time span of this revision (1 INSC96R graft and 2 INS corrected grafts). The results are in line with the previous data, showing non-significantly lower level of proliferation at 1 mo by Ki67 staining. Generation of mice transplanted for 6 months would require 8-9 months, which is out of the scope of this revision.

Complementary to the KI67 proliferation analysis presented in Figure 6E, we have now performed immunostainings for a different proliferation marker, PCNA, across time points and genotypes. Results are presented in Figure 6F. The quantification of PCNA^+^ INS+ cells showed similar trend of proliferation as with KI67, strengthening this part of the data. At 1 mo, the PCNA labelling is significantly lower in the mutant grafts.

4) The authors should acknowledge that the β-like cells are not functional in vitro and in vivo, they do not respond with increases in insulin secretion to a glucose challenge (also depolarization shows only modest secretion). It therefore impossible to draw strong conclusions as to the effects of the mutations on mature β cell physiology. The authors are encouraged to discuss this limitation.

We agree with the reviewers that the functionality of the stem cell derived β-cells is not comparable to mature β-cells, we have discussed this limitation of the β-like cells generated with the current differentiation protocols in the Discussion (first paragraph).